# Therapeutic Agent-Loaded Fibrous Scaffolds for Biomedical Applications

**DOI:** 10.3390/pharmaceutics15051522

**Published:** 2023-05-17

**Authors:** Dongsik Park, Su Jin Lee, Dong Kyu Choi, Jee-Woong Park

**Affiliations:** 1Drug Manufacturing Center, Daegu-Gyeongbuk Medical Innovation Foundation (K-MEDI Hub), Daegu 41061, Republic of Korea; 2New Drug Development Center (NDDC), Daegu-Gyeongbuk Medical Innovation Foundation (K-MEDI Hub), Daegu 41061, Republic of Korea; 3Medical Device Development Center, Daegu-Gyeongbuk Medical Innovation Foundation (K-MEDI Hub), Daegu 41061, Republic of Korea

**Keywords:** fiber scaffolds, drug carrier, electrospinning, 3D bioprinting

## Abstract

Tissue engineering is a sophisticated field that involves the integration of various disciplines, such as clinical medicine, material science, and life science, to repair or regenerate damaged tissues and organs. To achieve the successful regeneration of damaged or diseased tissues, it is necessary to fabricate biomimetic scaffolds that provide structural support to the surrounding cells and tissues. Fibrous scaffolds loaded with therapeutic agents have shown considerable potential in tissue engineering. In this comprehensive review, we examine various methods for fabricating bioactive molecule-loaded fibrous scaffolds, including preparation methods for fibrous scaffolds and drug-loading techniques. Additionally, we delved into the recent biomedical applications of these scaffolds, such as tissue regeneration, inhibition of tumor recurrence, and immunomodulation. The aim of this review is to discuss the latest research trends in fibrous scaffold manufacturing methods, materials, drug-loading methods with parameter information, and therapeutic applications with the goal of contributing to the development of new technologies or improvements to existing ones.

## 1. Introduction

Fibrous scaffolds exhibit excellent characteristics, including adjustable biodegradability, appropriate mechanical properties, and high porosity, which make them promising materials for use in regenerative medicine [1,2,3]. In addition to providing a suitable environment for cell growth [4], these scaffolds offer flexibility for drug loading [5], enabling them to be used as implantable [6] and degradable drug carriers [7] for treating diseases and repairing tissues. The local controllable release feature of these fibrous scaffolds further enhances their potential for targeted drug delivery, making them a promising option for a range of biomedical applications.

Natural materials are widely used to fabricate fibrous scaffolds for tissue engineering and regenerative medicine because of their biocompatibility and ability to mimic the extracellular matrix (ECM) of natural tissues [8,9]. Among the various natural materials, polysaccharides [10], alginate [11], and proteins such as collagen [12] have been widely explored for the fabrication of fibrous scaffolds. These natural materials are biocompatible and are less likely to cause an immune response or toxicity in the body, making them suitable for use in tissue engineering and regenerative medicine. Polysaccharides and proteins are the main components of ECM in natural tissues, and their use in scaffolds can provide a similar microenvironment to support cell attachment, proliferation, and differentiation. Moreover, the physical and chemical properties of natural materials can be tailored to fabricate a scaffold with specific properties, such as porosity, stiffness, and degradation rate, which can influence cell behavior and tissue regeneration.

Among the several methods available for fabricating nanofibrous membranes, electrospinning shows significant potential because it allows the incorporation of drugs into the membrane, which can then be released at a controlled rate over different time periods [13]. Electrospinning is a simple yet effective technique for producing fibrous materials in the micrometer to nanoscale range and can be utilized in various applications such as tissue regeneration, drug delivery, and wound dressing. Three-dimensional (3D) printing technology allows the manipulation of individual 3D shapes with high precision, including the fabrication of fiber scaffolds [7]. This approach has several advantages, such as the ability to produce complex, customized, and patient-specific designs, as well as precise control over properties such as pore size, interconnectivity, and mechanical characteristics.

Microfluidic technologies have shown promise for producing micro- and nanoscale fiber structures for a range of applications without requiring complex equipment or precise control over microflows [14].

To achieve a desired therapeutic effect, drugs or bioactive molecules can be incorporated into fibrous scaffolds via a process known as drug loading [15]. When selecting a fiber matrix for drug loading, it is crucial to consider the interactions between the drug and the fiber scaffolds. Covalent bonds, hydrogen bonds, and electrostatic interactions can be used to link the drug to the fibers.

In this review, natural fiber materials, methods for fabricating fibrous scaffolds, loading methods, and state-of-the-art applications of fabricated fibrous scaffolds are discussed.

## 2. Natural Materials for Fibrous Scaffolds

Tissue-engineering scaffolds are designed to temporarily replace the native ECM, thus providing mechanical support for cells and regulating cell behavior. The ECM is primarily composed of collagen, elastin, glycoproteins, and polysaccharides; therefore, natural polymers, such as proteins and polysaccharides, have been extensively studied for scaffold production using various methods.

### 2.1. Polysaccharides

Natural polysaccharides such as arginate, chitosan, and hyaluronic acid are highly valued in biomedical science owing to their outstanding properties [9]. (Figure 1) Compared to other polymers, they are easily customizable, biocompatible, bioactive, homogeneous, and possess bioadhesive properties. Moreover, they can replicate the ECM microenvironment, making them superior to other materials. Polysaccharides are natural polymers that have been extensively studied for the production of fibrous scaffolds in tissue engineering. One of their advantages is that they are naturally occurring molecules in the human body, making them well-tolerated by host tissues and reducing the likelihood of an immune response or adverse reactions. In addition, polysaccharides are often biodegradable, indicating that they can be safely broken down by enzymes in the body. The abundance and diversity of polysaccharides, which are available from various sources such as plants, animals, and microorganisms, allow the production of scaffolds with unique mechanical, chemical, and biological properties [16]. All these factors make polysaccharides promising materials for scaffold production in tissue engineering.

#### 2.1.1. Alginate

Alginate is a natural polymer derived from brown seaweed that is widely used in tissue engineering because of its unique properties. Alginate is composed of two types of monomers, guluronic acid and mannuronic acid, arranged in different ratios, depending on the seaweed source [22] (Figure 1a). One advantage of alginate as a natural material for fibrous scaffolds is its biocompatibility and non-toxicity, as it is naturally derived and does not cause an immune response [23]. Alginate hydrogels can also be easily modified with various chemical groups or bioactive molecules to improve their properties and mimic the ECM. Alginate-based biomaterials can be further modified by combining them with other biomaterials, immobilizing specific ligands (such as peptides and sugar molecules), and subjecting them to crosslinking treatments. These modifications allow for the fine-tuning of the structural and functional properties of alginate, including its biodegradability, mechanical strength, gelation behavior, and cell interactions [24]. Alginate-based biomaterials can potentially function as drug delivery vehicles [25] and cell carriers [26] in tissue-engineering applications.

#### 2.1.2. Chitosan

Chitosan, a natural polymer derived from chitin (Figure 1b), is often used to fabricate fibrous scaffolds for tissue-engineering applications. Its unique properties such as biocompatibility, biodegradability, and antimicrobial activity make it a promising candidate for tissue engineering applications [27]. Chitosan can be processed into fibrous scaffolds using various techniques such as electrospinning [28] and gas foaming [29], which results in a 3D structure that mimics the ECM tissues. Moreover, chitosan can form blends with other polymers to form scaffolds with the desired properties. The cationic nature of chitosan enables it to form polyelectrolyte complexes with anionic glycosaminoglycans, which have been shown to regulate the activity of growth factors and cytokines, making them suitable for bone and cartilage regeneration [30]. Additionally, the degradation rate of chitosan can be adjusted by modifying its degree of deacetylation, enabling it to satisfy the regenerative requirements of different tissues [31]. The functional groups on chitosan also facilitate conjugation with biologically active molecules, such as adhesive proteins and peptides, to support cell growth, differentiation, and proliferation in various fields, including dentistry, ophthalmology, veterinary science, biomedicine, the drink industry, and biotechnology [32].

#### 2.1.3. Hyaluronic Acid

Hyaluronic acid (HA) is a polysaccharide naturally found in the body (Figure 1c), particularly in the ECM of connective tissues, including the skin, cartilage, and connective tissue. It exhibits good biocompatibility and minimizes the immune response when implanted in the body. HA is degraded by hyaluronidase, which cleaves the glycosidic bonds between sugar molecules [33]. Hyaluronidase enzymes are present in various tissues of the body, including the liver, kidney, spleen, and lymph nodes, and degraded fragments can be metabolized and eliminated from the body through the lymphatic system.

Low-molecular-weight HA has been found to promote the proliferation and migration of vascular endothelial cells. Moreover, it has been reported that low-molecular-weight HA can stimulate the expression of signaling molecules such as ezrin, which is a crucial protein for cellular adhesion [33]. Furthermore, HA exhibits antioxidant properties, and it is hypothesized that high-molecular-weight HA may protect against the effects of reactive oxygen species. HA possesses antioxidative properties such as the ability to reduce ultraviolet (UV) light-induced apoptosis and acid-induced DNA damage [34]. These properties make HA an attractive candidate for use in various fibrous scaffolds, including wound healing, cartilage repair, and tissue regeneration.

### 2.2. Proteins

Proteins are naturally biodegradable materials that have garnered significant attention in the field of tissue engineering for scaffold production. Their remarkable biocompatibility and biodegradability, coupled with their degradation products and amino acids, make them prime candidates for nutrient resorption. Moreover, some proteins induce minimal tissue inflammatory responses [35].

Among proteins, fibrous proteins such as collagen are of particular interest for scaffold production because of their unique properties. Collagen is a natural component of the ECM in many tissues, including skin, bone, and cartilage [12]. This makes it highly biocompatible and facilitates cellular attachment and proliferation, making it an ideal material for tissue-engineering applications. These fibrous proteins exhibit highly repetitive amino acid sequences that primarily serve mechanical and architectural functions. Repetitive primary sequences give rise to relatively uniform secondary structures through self-assembly, resulting in the formation of fibrous structural materials. Such structures offer several benefits, including high mechanical strength, stability, and enhanced cellular attachment and proliferation, all of which are critical for tissue regeneration and repair [36].

#### 2.2.1. Collagen

Collagen, an abundant protein in the human body (Figure 2), is a highly investigated biomolecule of the ECM because of its wide distribution in connective tissues and superior mechanical properties, which make it an ideal material for load-bearing tissues such as bones and tendons [37]. Collagen is a useful material for applications in drug delivery systems and tissue engineering, owing to its tensile strength, biomimetic nature, low antigenicity, and safety profile [38]. Collagen is biocompatible and is degraded by human collagenases. Moreover, cell–collagen interactions are naturally facilitated as specific peptide sequences within collagen molecules are recognized by cell receptors [39]. Therefore, collagen can support the growth and differentiation of various cell types, making it a versatile material for various tissue-engineering applications. In addition, the mechanical strength and degradation properties of collagen-based scaffolds can be controlled using various chemical, physical, and enzymatic treatments.

Furthermore, collagen can be easily extracted from natural sources such as bovine hide, pig skin, and human placenta, making it a readily available and cost-effective material for scaffold production [40]. In addition, marine organisms are being increasingly investigated as a potential source of collagen due to a lower risk of transmissible diseases [41]. Unfortunately, the fishing industry discards a significant amount of marine biomass, up to 85%, leading to environmental concerns [42]. However, extracting collagen from these wastes can offer both environmental and economic benefits. Marine collagens have unique properties, such as biocompatibility, reduced zoonotic and immunological risks, and fewer religious restrictions. Finally, collagen can be processed into various forms such as gels, fibers, and films, allowing for the fabrication of scaffolds with different properties and structures to suit specific tissue-engineering needs.

**Figure 2 pharmaceutics-15-01522-f002:**
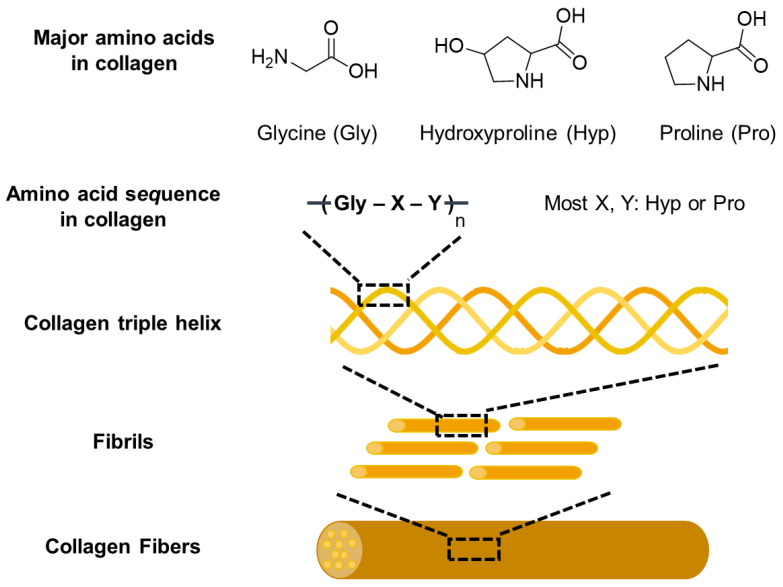
Structure of collagen [43,44,45,46,47].

#### 2.2.2. Silk Fibroin

Silk fibroin (SF) is a natural protein-based polymer that has attracted significant attention as a material for fibrous scaffolds in tissue engineering and regenerative medicine. Silkworms are commonly used in silk research and the most common silk originates from *Bombyx mori*, a mulberry-feeding silkworm [48]. Silkworm cocoons consist of SF, a load-bearing protein, and sericin, a gumming agent. Sericin is often removed from SF to ensure biocompatibility in tissue-engineering applications. The sodium carbonate degumming is the most commonly used method for sericin removal [49]. However, after degumming, the average diameter of silk fibers is reduced to 10–25 µm. Researchers have designed chimeric recombinant spider silk proteins that can produce large quantities of artificial spider silk in bacterial shake-flask culture.

SF is composed of heavy (H) and light (L) chains linked by disulfide bonds along with a glycoprotein called P25 [6] (Figure 3). The H-chain, L-chain, and P25 are three polypeptides that form the cocoon of *Bombyx mori* at a molar ratio of 6:6:1 [50]. The H-chain is rich in glycine and alanine with a repeating Gly-X dipeptide motif, including two hexapeptides with specific peptide sequences. SF has two crystalline structures, silk I and silk II, with silk II being the most stable owing to its strong hydrogen bonding [51]. SF also exists in an amorphous state with α-helices, turns, and random-coil structures. The unstable crystal structure of SF is silk III, which exists at the air–water interface of the regenerated SF solutions.

SF fibers from the *Bombyx mori* cocoons have exceptional mechanical properties, including large break strain, ultimate strength, and toughness, surpassing those of natural and synthetic fibers [52]. SF has been used as a scaffold material in load-bearing tissue-engineering applications, particularly in musculoskeletal tissue engineering [53]. However, SF scaffolds fabricated from regenerated SF (RSF) solutions are typically brittle and weak compared with raw SF fibers because of the lack of hierarchical and secondary structures [54]. Various strategies such as crosslinking, porogens [55], and 3D bioprinting [56] have been used to improve the mechanical properties of RSF-produced silk scaffolds, resulting in scaffolds with mechanical properties similar to those of the native tissue being repaired.

SF is a biologically inert and biocompatible natural polymer approved by the U.S. Food and Drug Administration (FDA) for use as a biomaterial in 1993 [57]. SF has been shown to have blood compatibility and has been used successfully in cell culture to guide bone regeneration [58]. In vitro and in vivo studies have demonstrated that there is no significant immune reaction or inflammatory response to SF films or fibers [59], and SF has been shown to be a suitable alternative to collagen in some applications.

**Figure 3 pharmaceutics-15-01522-f003:**
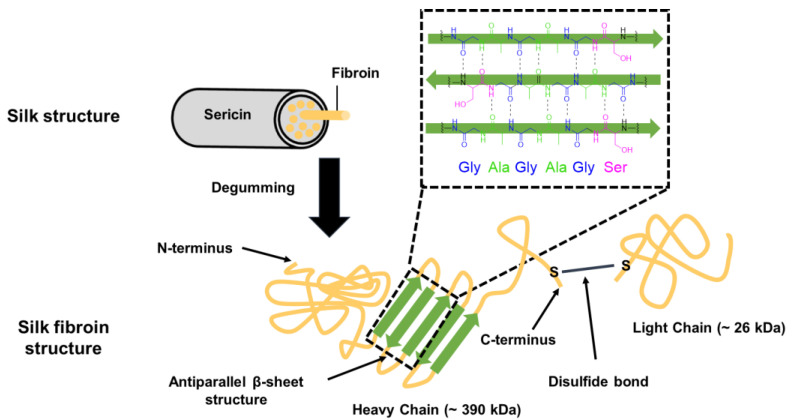
Structural illustration of silk fibroin [60,61,62,63,64].

#### 2.2.3. Elastin

Elastin is a fibrous protein that is predominantly found in the elastic tissues of the body, such as the skin, lung tissue, and blood vessels. Elastogenesis, which primarily occurs during the late fetal and early neonatal periods, is regulated by various cell types, including smooth muscle cells, fibroblasts, endothelial cells, chondroblasts, and mesothelial cells [65]. It is highly elastic and can stretch and recoil without deformation. Elastin-based scaffolds have been used in tissue-engineering applications such as vascular grafts and skin substitutes.

Tropoelastin is a water-soluble protein that serves as the precursor to the crosslinked elastin. It is composed of alternating hydrophobic and hydrophilic domains and has a molecular weight of approximately 72 kDa (Figure 4). After secretion into the ECM, tropoelastin interacts with microfibrillar scaffolds to facilitate elastic fiber formation [66]. Crosslinking is accomplished through an entropically driven process of coacervation, which causes tropoelastin molecules to aggregate with increasing temperature. This process is closely regulated to match the physiological conditions. Finally, copper-dependent lysyl oxidase is responsible for the crosslinking of tropoelastin [67]. Elastin can withstand billions of extension and recoil cycles without failure [68].

Various techniques have been explored to produce elastin-based molecules owing to the significance of elastin in tissues. These approaches include the extraction and purification of elastin from animal sources, expression of tropoelastin using recombinant methods, and manipulation of elastin-like peptides through engineering.

Elastin can be obtained from animal sources through the partial hydrolysis of the peptide chain using agents such as hot alkali, guanidine, oxalic acid, and potassium hydroxide [68]. Various methods yield different versions of soluble elastin with similar frameworks, including the removal of other macromolecules and fragmentation of elastin by cleaving peptide bonds [69]. The fragmented peptides possessed properties similar to those of tropoelastin, including the capacity to coacervate. However, the final product obtained from animal sources is a heterogeneous mixture of partially crosslinked elastin peptides with compromised biological properties because of the degradation and high levels of contaminants, depending on the protocol used [70].

Researchers have explored various methods for producing tropoelastin [71]. One approach involves obtaining tropoelastin from animal sources at the late fetal or neonatal stage before it is crosslinked into elastin by feeding newborn animals a copper-deficient diet, which is a critical cofactor for lysyl oxidase activity [72]. However, the extracted tropoelastin is more susceptible to degradation by proteases and has an overall positive charge owing to the presence of uncrosslinked lysine residues.

Recombinant human tropoelastin using *Escherichia coli* (*E. coli*) has been used as a model to study the behavior of human elastin and fabricated into tropoelastin-based biomaterials for tissue-engineering applications [73]. However, the extensive purification of proteins from bacterial systems is required. Elastin-like peptides (ELPs) have been studied as alternatives [74]. ELPs are synthesized by modifying the amino acid sequence, side chains, and synthesis, and can be used to control the chemical and biological properties of the material. ELPs have been shown to elicit minimal inflammatory responses when implanted in vivo and the addition of a cell-binding sequence to the ELP chain promotes cell adhesion, proliferation, and migration [75].

**Figure 4 pharmaceutics-15-01522-f004:**
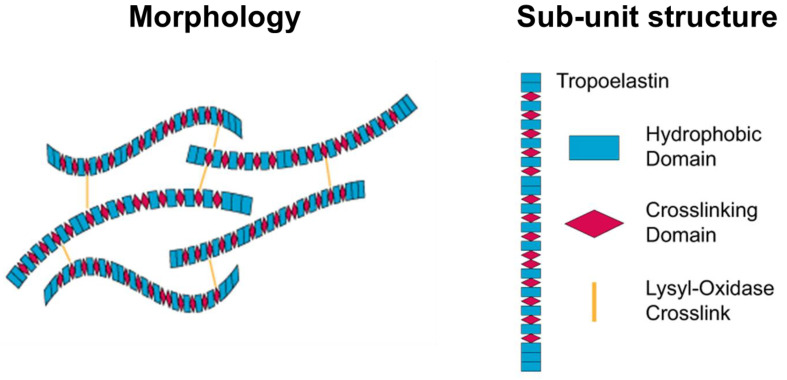
Structural illustration of elastin. (Reprinted with permission from ref. [76]. Copyright 2019 Springer Nature).

## 3. Fabrication Methods

Various methods, including electrospinning, 3D printing, and microfluidics, have been employed to fabricate fibrous scaffolds. Electrospinning is a versatile and widely used technique for producing fine fibers from polymer solutions or melts. Three-dimensional printing allows for the precise control over of scaffold geometry and ability to incorporate multiple materials and functional gradients. Three-dimensional printing can also be used to produce scaffolds with high mechanical strength and structural integrity. Microfluidics uses microscale channels and chambers to control the flow of fluids and deposition of fibers. It offers the advantages of high-throughput fabrication and the ability to create complex scaffold architectures. This section aims to provide a summary of the methods for the fabrication of fibrous scaffolds with a detailed classification and principles of the mentioned technologies, along with recent research examples.

### 3.1. Electrospinning

Over the past few decades, electrospinning has become a crucial method for fabricating 3D fibrous scaffolds with fiber diameters ranging from nanometers to micrometers. This technique is the most extensively studied and widely used method for scaffold fabrication. Electrospinning utilizes a high electric voltage to synthesize nanofibers, and a typical electrospinning setup consists of a syringe pump, power supply, metallic needle, and metallic collector [13]. The electric field causes the polymeric solution to flow in the direction of the electric field and generates ultrafine fibers from the tip of the metallic needle, which are collected on the collector drum. The diameters of the resulting fibers range from nanometers to micrometers and are called electrospun micro- and nanofibers. To achieve uniformly sized and beadless fibers with well-developed inter- and intra-porous morphology of the scaffold, certain factors need to be optimized, including the applied voltage, solution flow rate, viscosity of the solution, concentration of the solution, distance between the needle and collector, and humidity of the electrospinning chamber. Fiber mats manufactured using this method possess large surface areas and controllable pores, making them potentially useful for diagnosing and treating cancer cells [77]. One limitation of electrospinning is that non-conductive polymers cannot be electrospun; however, this limitation can be overcome by adding appropriate conductive additives to the polymer solution. Various techniques have been developed to produce complex nanofibers using electrospinning methods [13,28,78] (Figure 5).

#### 3.1.1. Bicomponent Spinning

Bicomponent spinning is used to fabricate scaffolds with two polymers to provide a particular combination of mechanical, chemical, or biological properties that are desired for the scaffold. The bicomponent spinning method uses a spinning nozzle with two or more separate openings to extrude two different polymer solutions simultaneously. The two solutions are arranged in a sheath–core configuration, with one polymer forming the core and the other forming the sheath. As the fibers are extruded, the two polymers combine to form a single composite fiber with a well-defined sheath–core structure.

Wang et al. developed bicomponent electrospun scaffolds for bone tissue regeneration by incorporating calcium phosphate (Ca-P) and bone morphogenetic protein 2 (BMP2) in separate fibers [79]. Bicomponent scaffolds were fabricated using a dual-source dual-power electrospinning technique, resulting in evenly distributed fibers. The bicomponent scaffolds exhibited enhanced cell proliferation, alkaline phosphatase activity, cell mineralization, and the gene expression of osteogenic markers compared to the monocomponent scaffolds because of the synergistic effect of BMP2 and Ca-P nanoparticles.

Liu et al. investigated the formation and properties of fibrous bicomponent scaffolds for the controlled dual delivery of glial cell line-derived neurotrophic factor (GDNF) and nerve growth factor (NGF) for peripheral nerve tissue regeneration [80]. Bicomponent scaffolds with various ratios of GDNF/lactic-co-glycolic acid (PLGA) and NGF/poly-d,l-lactide (PDLLA) fibers were fabricated using bicomponent electrospinning techniques. The release profiles of GDNF and NGF were controlled, and a concurrent and sustained release from the bicomponent scaffolds was achieved. The released GDNF and NGF exhibited a synergistic effect in promoting neural differentiation. Bicomponent scaffolds are promising for use in tissue engineering for peripheral nerve regeneration.

#### 3.1.2. Coaxial Electrospinning

The coaxial electrospinning process allows the synthesis of diverse nanofibers with multiple types of polymer solutions as well as non-polymeric materials without filament-forming properties, such as ceramics and semiconducting materials. Coaxial electrospinning uses a setup similar to that of standard electrospinning but with an ejection device that consists of concentric needles. As the solutions are ejected from these needles, core–sheath fibers are formed [81]. Coaxial electrospinning has been modified to fabricate more layers [82] and multichannel structures, expanding its applications to various fields, such as biomedicine, electrochemistry, and environmental chemistry. However, controlling the interfacial behavior between adjacent polymeric liquids is complex because it involves the control of multiple parameters. Identifying the specific process parameters is important for realizing useful one-dimensional nanostructures for various applications.

Coaxial electrospinning involves the use of two or more solutions, with one solution encapsulated within the other, resulting in a core–shell structure. In contrast, bicomponent electrospinning involves spinning two or more solutions side-by-side, resulting in side-by-side structures.

Reise et al. fabricated poly(L-lactide-co-D,L-lactide) fiber mats with different distributions of metronidazole using coaxial and conventional electrospinning techniques [83]. The coaxially fabricated fibers exhibited improved and sustained drug release compared with conventional fibers and had a higher antibacterial effect and better biocompatibility with human gingival fibroblasts. This study suggests that coaxially fabricated fibers have potential as drug delivery systems for the treatment of local periodontitis.

#### 3.1.3. Multijet Electrospinning

Multijet electrospinning is a technique that involves the use of multiple needles or nozzles to generate multiple jets of polymer solution or melt [83]. Multijet electrospinning allows the deposition of multiple fibers of different materials and sizes on the same substrate, thus enabling the fabrication of multilayered structures or composite materials. This technique offers increased production rates, the ability to control fiber morphology and diameter, and the production of fibers with various chemical and physical properties [84,85].

We et al. examined the impact of various solution parameters such as the dielectric constant, conductivity, polarity, and surface tension on the formation of multiple jets during electrospinning [84]. The results showed that solutions with higher dielectric constants and larger surface tensions were more likely to generate two to six jets with shorter stable jet lengths under low voltages. This study provides insight into the jet evolution process and its relationship with the mass production of nanofibers during electrospinning.

#### 3.1.4. Emulsion Electrospinning

Emulsion electrospinning is a novel and simple method for fabricating core–shell nanofibers. This technique can utilize either a water-in-oil (W/O) or oil-in-water (O/W) emulsion to produce core–shell fibers that directly encapsulate hydrophilic or hydrophobic substances, respectively. This method eliminates the need for organic solvents, which are heavily regulated in food applications. Emulsion electrospun fibers can encapsulate a variety of food ingredients such as vitamins, carotenoids, polyphenols, enzymes, peptides, oils, flavors, and probiotics, including hydrophilic and hydrophobic compounds [86].

Norouzi et al. fabricated sodium alginate/poly(ε-caprolactone) (PCL) core–shell nanofibers using water-in-oil emulsion electrospinning for potential use as drug vehicles and tissue-engineering scaffolds [87].

Emulsion electrospinning can be used to fabricate biofunctional tissue-engineered scaffolds. Xiaoqiang et al. used emulsion electrospinning to develop biodegradable fibrous mats with encapsulated human NGF [88]. The NGF protein was effectively encapsulated and released in a sustained manner. The bioactivity of NGF released from the fibrous mats was confirmed by testing the neurite outgrowth of rat pheochromocytoma cells (PC12).

The main features of all the electrospinning methods are listed in Table 1.

### 3.2. Three-Dimensional Printing

Manipulation of individual 3D shapes with high precision can be achieved using 3D printing technology. Three-dimensional printing-based fiber scaffold fabrication has several advantages, including the ability to create complex, customized, and patient-specific designs, as well as precise control over pore size, interconnectivity, and mechanical properties. Three-dimensional printing can be categorized based on the specific techniques employed in the process. These techniques include fused deposition modeling (FDM), stereolithography (SLA), selective laser sintering (SLS), and selective laser melting (SLM) (Figure 6).

#### 3.2.1. Fused Deposition Modeling

FDM is a commonly used additive technique in the manufacturing industry. It involves the use of molten thermoplastic polymers such as acrylonitrile butadiene styrene and polylactic acid, which are heated above their glass transition temperatures and sequentially sprayed layer-by-layer to build the desired 3D model. The speed of plotting, rate of filament given, and thickness of the layer are the key factors affecting the FDM performance. Researchers have explored the potential of FDM for drug delivery and drug-eluting implantation devices. For instance, Stewart et al. developed an implantable system that employed 3D printing technology to deliver diverse samples [89], and Giri et al. proposed a novel approach for generating gastro-retentive floating tablets with zero-order drug release patterns [90]. These studies highlight the potential of FDM as an efficient and cost-effective method for developing controlled-release dosage forms with improved release rates.

#### 3.2.2. Stereolithography

The SLA technique utilizes a specialized photopolymer that undergoes a chemical reaction when exposed to UV/IR light, resulting in alterations in its physicochemical properties. Despite its advantages, such as excellent surface texture and the fastest and finest resolution 3D printers, this technique has limitations such as brittleness, low impact strength, and limited lifespan owing to the degradation of physical characteristics with time. Xu et al. utilized SLA 3D printing to manufacture novel indwelling bladder devices using an elastic polymer for the extended and localized delivery of lidocaine hydrochloride [91]. In vitro studies demonstrated acceptable blood compatibility, good resistance to compressive and stretching forces, and a complete release of lidocaine within 4 days for hollow devices and sustained release for up to 14 days for solid devices.

However, the authors reported unforeseen drug–polymer reactions in SLA 3D printing in another article [92]. This underscores the importance of screening for photoreactive monomers to ensure the compatibility of drug-loaded oral dosage forms produced through SLA.

#### 3.2.3. Selective Laser Sintering

In SLS, a laser is used to melt and harden polymer mixtures for fabricating 3D objects. SLS printers use a CO_2_ laser for high-resolution models at room temperature. However, the mutagenic properties of the polymer limit their use. By adjusting the SLS process parameters, users can have excellent control over the microstructures of the fabricated scaffolds [93]. The parameters used in SLS, such as temperature and laser momentum, have a significant impact on the performance and stability of the final product.

Du et al. fabricated a bioinspired osteochondral scaffold using SLS with PCL and hydroxyapatite/PCL microspheres [94]. These scaffolds demonstrated biocompatibility and supported cell adhesion and proliferation in vitro. The scaffolds were implanted into rabbit models to evaluate their repair effects on osteochondral defects. The multilayer scaffolds induced articular cartilage formation, accelerated subchondral bone regeneration, and integrated well with native tissues. This study suggests that SLS can be used to fabricate well-designed multilayer scaffolds for osteochondral repair.

Thakkar et al. investigated the effect of processing parameters and composition on drug degradation, crystallinity, and quality of SLS-based 3D-printed dosage forms using a light-sensitive drug, nifedipine [95]. The results showed that formulation and processing parameters have a critical impact on stability and performance. Candurin and laser speed had a strong negative correlation with drug degradation, hardness, and weight, whereas surface temperature had a strong positive correlation with drug degradation, amorphous conversion, and hardness.

#### 3.2.4. Selective Laser Melting

SLM is a type of powder bed fusion technique that involves using a bed of powder granules with a specified density. Heat from a thermal source is used to fuse the particles and regulate the fusion process. Laser light is projected onto the powder bed, generating heat that melts the construction material. The molten material solidifies as it cools to form the desired object. A portion of the bed remains in the unmelted form to support the object. The unused powder bed is removed after the object is fully formed.

Shuai et al. used SLM to fabricate an iron–manganese (Fe-Mn) biodegradable scaffold [96]. The scaffold exhibited suitable mechanical properties including yield strength and good cytocompatibility. This study concluded that the SLM-processed Fe-Mn scaffold is a promising material for bone repair.

Table 2 outlines the advantages and disadvantages of each method discussed above. Despite these limitations, 3D printing is promising for the fabrication of fibrous scaffolds. In addition to the abovementioned 3D printing methods (FDM, SLA, SLS, and SLM), other methods such as EHD printing [97] and inkjet printing [98] have been utilized in 3D-printing applications.

### 3.3. Microfluidics

Microfluidic technologies have recently demonstrated considerable potential for producing microscale and nanoscale fiber structures for various applications without the need for complicated devices and with excellent manipulation of microflows [14]. In this section, the principles of microfluidic systems are discussed, followed by an explanation of the microfluidic strategies employed for fiber production from the perspective of their platforms and geometries.

#### 3.3.1. Principle of the Microfluidic Synthesis of Fiber

Microfluidic systems typically operate under laminar flow conditions (i.e., at low Reynolds numbers), providing superior control over the synthesis process. Therefore, the structure and functionality of the produced fibers can be finely tuned by manipulating the flows in microchannels, which are affected by surface tension and fluid viscosity and are mixed only through diffusion at the interface between flows under laminar flow conditions [99]. An example of the glass capillary used for the synthesis of the fiber is shown in Figure 7.

During a typical microfluidic fiber fabrication process, sample fluids containing polymer precursors and potentially non-polymerizable sheath solutions are introduced into separate input ports of the microchannels. Under this condition, core–sheath flow is generated. In this fluidic channel, the outer liquid plays a crucial role in carrying the inner liquid, shaping the geometry of the inner fluid, preventing contact with the channel walls, and avoiding channel clogging after fiber formation [101]. By controlling the flow rates of various fluids and the dimensions, shapes, and arrangements of the microchannels, the size and shape of the final fibers can be precisely modulated. Another important part of the fabrication of fibers is the fluid solidification process, which must be sufficiently fast to control and restore the desired fiber shapes [102].

Another factor to consider is the behavior of non-Newtonian fluids [103]. At the microscale, the fluid flow can be influenced by a variety of factors that are not significant at larger scales. These factors include the confinement of the fluid to small channels, presence of solid walls that can affect the flow, and influence of surface tension and viscous dissipation. Due to these factors, microfluidic fluids can exhibit non-Newtonian behavior, implying that their viscosity is not constant but instead depends on the shear rate or applied stress.

One of the most common types of non-Newtonian behaviors observed in microfluidic fluids is shear thinning, in which the viscosity of the fluid decreases as the shear rate increases [104]. This behavior is typically attributed to the deformation of the fluid microstructure under shear, such as the alignment of particles or stretching of polymer chains. Another type of non-Newtonian behavior observed in microfluidic fluids is viscoelasticity [105], in which the fluid exhibits both viscous and elastic properties. Viscoelasticity arises from the ability of the fluid to store and release energy when deformed, such as when a polymer chain stretches and then recoils. This behavior can lead to complex flow phenomena such as flow instability and vortex formation in microfluidic channels. Microfluidic fluids that exhibit viscoelastic behavior include polymer, protein, and surfactant solutions. These factors must be considered when designing microchannels for the fabrication of fibrous scaffolds.

#### 3.3.2. Microfluidic Devices

Microfluidic devices developed for fiber production typically utilize microchannels made from cylindrical or rectangular shapes using materials such as polydimethylsiloxane (PDMS) [106], cyclic olefin copolymers [107], poly(methyl methacrylate) (PMMA) [108], and glass.

Soft lithography using elastomeric materials such as PDMS or computer numerical control/milling machines with materials such as PMMA, fluoropolymers such as polytetrafluoroethylene (PTFE), or metals can be used to fabricate two-dimensional microfluidic chips. These chips have numerous benefits such as high reproducibility, compatibility with bio/chemical components, low production costs, microchannels with precise designs, and rapid prototyping. While the conventional soft lithography technique mostly results in microchannels with rectangular cross-sections, some researchers have proposed alternative methods to generate channels with different cross-sectional shapes. In contrast, the fabrication process for microfluidic devices using a master mold fabricated via soft lithography is relatively affordable. However, this method requires an initial investment in cleanroom facilities.

Wang et al. developed a microfluidic technology to generate islet-laden fibers in a single step to protect them from immune responses [109]. Islet-laden fibers with mesenchymal stromal cells (MSCs) transplanted into diabetic rats restored normoglycemia for an extended period and improved the glucose-stimulated response of the islets. Fibers including islets and MSCs provide a versatile platform for improving cell preservation and function after in vivo transplantation.

Nguyen et al. presented a novel technique for producing hollow fibers using a triple-flow microfluidic device made of PDMS [110]. Alginate, induced with CaCl_2_, was used to form hollow fibers in a coaxial flow system. Two PDMS replicas with a semicylindrical microchannel were combined to create a circular cross-section and produce mild and continuous coaxial flows for fiber fabrication without the need for complex glass microcapillaries. The resulting hollow fibers were flexible, mechanically strong, permeable, and biocompatible, making them suitable scaffolds for the attachment and proliferation of human umbilical vein endothelial cells (HUVECs) to mimic blood vessels.

Regarding glass materials, researchers have utilized the advantages of glass to fabricate microfluidic devices with diverse structures by using a pipette puller and cutting to different orifice diameters. The benefits of glass materials include hydrophilicity, circular cross-sections to form a stable coaxial flow, and easy surfaces. Although glass capillary-based systems are relatively inexpensive and rapid, they require labor-intensive and time-consuming procedures, specialized tools, such as microforges or microcapillary pullers, and skilled personnel.

Wang et al. developed a microfluidic 3D printing technique using a glass capillary [111] for the fabrication of fibrous scaffolds incorporating black phosphorus and photothermally responsive channels, which could enhance vascularization and bone regeneration. Scaffolds are promising candidates for tissue and vascular ingrowth in various tissue-engineering applications.

Microfluidic systems offer a wide range of benefits including energy efficiency, cost-effectiveness, precise controllability, and consistent production. These advantages enable the production of micro/nanoscale fibers with diverse structures and functionalities [112] as well as allow spatiotemporal control over the safe loading of multiple components owing to system flexibility.

### 3.4. Solution Blowing

Solution blowing utilizes compressed air as the driving force for fiber formation [113,114,115] (Figure 8). The most commonly used model is a concentric nozzle system, in which a polymer solution is pumped into the inner nozzle and a high-pressure gas is delivered through the outer nozzle. The resulting polymer solution leaves the nozzle as multiple strands and is stretched by the high-speed gas. High shear forces at the gas–solution interface cause solvent evaporation and the attenuation of the blown jet.

Li et al. developed a needleless blow-spinning system that utilizes a roll-to-roll nylon thread to deliver a spinning solution, combined with a vertically blowing airflow to draw high-quality nanofibers with a high throughput [114]. The system produced a wide range of nanofibers, including polymers, carbon, ceramics, and composites, with adjustable diameters at ultrahigh rates. The system can be upgraded from a single thread to multiple parallel threads and meshes, allowing the production of nanofibers on the kilogram scale without compromising their quality.

### 3.5. Centrifugal Spinning

Centrifugal spinning offers advantages over electrospinning in terms of production rate, cost, and safety [116,117,118]. This process involves feeding a polymer solution or melting it in a rotating chamber with multiple orifices. As the chamber rotates, the centrifugal force pushes the solution/melt toward the orifices. When the force exceeds the surface tension and viscosity, the polymer jet exits the orifices and is elongated as the solvent evaporates. The resulting nanofibers are collected in the collector plane. Centrifugal spinning is a promising technique for large-scale nanofiber production.

Mary et al. developed fibrous structures using the centrifugal spinning technique for tissue-engineered implantable materials and wound-dressing applications [116] because the centrifugal method is relatively simple and cost-effective for producing nanofibers. It can also be easily scaled up for large-scale production, rendering it suitable for industrial applications. In vitro cell line studies showed that the nanofiber mat was highly biocompatible and supported cell proliferation.

Martin-Alonso et al. developed multifunctional fibers by incorporating functionalized silver nanoparticles into poly acetic acid solutions by centrifugal force spinning [117]. The authors tested the suitability of the centrifugal force spinning technique for producing shape-memory fiber mats and found that the nanoparticle concentration of 2 wt% resulted in a good thermally activated shape-memory effect with high fixity and recovery ratios.

The parameters that should be considered for each fabrication method are listed in Table 3.

## 4. Drug-Loading Methods

Drug loading involves the incorporation of drugs or other bioactive molecules into a scaffold to achieve desired therapeutic effects. When selecting a fiber matrix for drug loading to achieve a specific drug release profile, it is essential to consider the interactions between the drug and the fiber scaffolds. In addition to hydrogen bonds and electrostatic interactions, covalent bonds can also be used to link the drug with fibers. The solubility of the drug in the polymer solution is another crucial factor to determine the optimal loading technique. However, the release process depends heavily on the drug-loading technique.

The composition, molecular weight, hydrophilicity, and degradation rate of the fiber polymer matrix affect the drug release behavior. Furthermore, release behavior is influenced by the relative molecular mass, crystallinity, solubility, and other properties of the drug. In this section, various strategies for loading drugs onto fibers are discussed, and a brief introduction to the methods for drug loading is illustrated in Figure 9.

### 4.1. Blending

The technique involving the blending of drugs with fibers and polymeric solutions remains crucial for drug loading and integration with fibers because it is the simplest and most cost-effective method, requiring only mixing and electrospinning [122,123]. The drug product is dispersed or dissolved in a polymer solution to produce encapsulated medicines by using a fabrication process for fibrous scaffolds. Specifically, drugs or functional nanoparticles with organic solvent resistance and chemical stability can be combined with polymers to form homogeneous solutions of fibrous scaffolds. The resulting microfibers or nanofibers contain one or more drugs and can be fabricated [124].

This method provides several advantages such as high drug loading, uniform distribution of drug molecules, and increased initial release. However, there are some limitations to this approach, such as the potential loss of bioactivity of the bioactive agents in the presence of solvents or electric fields [125] and the aggregation of drug, leading to an initial burst release that results in toxicity and reduces the effective lifespan of the medical device.

Electrospun halloysite clay nanotubes were investigated as drug containers with poly(caprolactone)/gelatin microfibers [126]. The nanotubes were aligned within the electrospun fibers, resulting in biocompatible halloysite-doped membranes. These membranes also demonstrated a sustained release of metronidazole for 20 days, indicating the potential use of halloysite clay nanotubes incorporated into electrospun membranes as drug containers for clinical applications.

Recently, a curcumin-loaded covalent organic framework (CUR@COF) was developed by condensation and the Schiff base reaction to establish pH-triggered drug release for wound dressings by incorporation into PCL nanofibrous membranes (CUR@COF/PCL NFMs) [127]. A pH-responsive CUR release profile was observed for the CUR@COF-based membrane, which had a high CUR-loading capacity. It was demonstrated that the CUR@COF/PCL NFM platform could accelerate wound healing and skin regeneration by controlling the expression of tumor necrosis factor-α (TNF-α) or vascular endothelial growth factor (VEGF) in an in vivo skin defect model, providing a novel application for COF-based drug/nanocomposites in wound healing.

### 4.2. Chemical Immobilization

Chemical treatment of the fiber surface for immobilization is an alternative strategy [128]. This enables the modification of the fiber adhesion properties, which can be further modified by the incorporation of pertinent functional groups, such as amines, carboxyls, hydroxyls, and thiols. Chemical conjugation techniques are better suited than physical adsorption onto the surfactant surface for accurately determining the quantity of drugs incorporated into the nanofiber mesh.

Electrospun nanocellulose and elastin hybrid scaffolds comprising gelatin/PDLLA, gelatin/PDLLA/nanocellulose, and gelatin/PDLLA/cellulose nanocrystal/elastin were fabricated [123]. The scaffolds exhibited a porous structure that persisted even after crosslinking. The scaffolds containing nanocellulose and elastin exhibited higher elongation at rupture than those containing pure gelatin/PDLLA, indicating that the composition could be modulated to tune the toughness of the scaffolds. The scaffolds were non-cytotoxic to fibroblasts in the extraction media and supported the attachment and proliferation of fibroblasts on their surfaces.

The surface of nanofibrillated bacterial cellulose (NFBC) was hydrophobized by surface modification with methyltrimethoxysilane (MTMS) in water [129]. Different NFBC: MTMS ratios were used to prepare cellulose samples with varying degrees of molar substitution and their structural characteristics and molecular dynamics were studied. This study suggests that surface-modified NFBC can be used as a filler in fiber-reinforced nanocomposite resins.

### 4.3. Physical Adsorption

Physical adsorption based on various forces, such as electrostatic, hydrophobic, hydrogen, and van der Waals forces and fiber–surface interactions, can be another strategy for drug loading onto fibrous scaffolds. Surface-modified pre-fabricated fibrous meshes possess a remarkably high surface-area-to-volume ratio, enabling drug-friendly physical immobilization on the surface and resulting in higher drug loading per unit mass compared with other devices [130]. However, non-specific adsorption may lead to the formation of weak bonds, causing large bursts of drug release. The rapid release of drugs from the nanofiber surface facilitates a facile management of the dosages of certain therapeutic agents, which should be considered for specialized applications.

Hydrophobic PCL nanofibers were modified with an amphiphilic triblock copolymer (PCL-b-PEG-b-PCL) to reinforce the physical hydrogel formed by the micellar crosslinking of the same copolymer [131]. The copolymer played a dual role in dispersing and stabilizing the nanofibers and in providing a framework for the hydrogel matrix. The addition of modified PCL nanofibers significantly enhanced the mechanical strength of the hydrogel, and the gel modulus could be tuned by varying the concentrations of the copolymer and nanofibers. The nanofiber-reinforced hydrogel has considerable potential as a new type of tissue sealant owing to its injectable and self-healing properties.

The anticancer drug camptothecin was physisorbed onto the surface of cellulose nanofibers [132] and the controlled release of camptothecin was studied. The results showed that enzyme-mediated cellulose nanofibers could be optimal carriers for controlled drug release formulations without any chemical excipients.

### 4.4. Emulsion Electrospinning

Bioactive factors such as growth factors, proteins, and nucleic acids must enter the cells to be effective; therefore, it is important to avoid contact with organic solvents and deliver them without inactivation [133]. Emulsion electrospinning can be used to partition bioactive substances into the aqueous phase, which enhances their activity. It also allows the simultaneous loading of multiple drugs and mitigates the explosive release of drugs.

Li et al. developed a local drug delivery system using an emulsion electrospun polymer patch with a core–sheath structure containing hydrophobic 10-hydroxycamptothecin (HCPT) and hydrophilic tea polyphenols (TPs) [134]. HCPT was used to suppress the growth of liver cancer cells, whereas TP was used to reduce the levels of free oxygen radicals and prevent the spread of cancer cells. The core–sheath structure allowed for the sustained and sequential release of the drugs, with HCPT being released first, followed by TP. This study demonstrated the potential benefits of using nanofiber membranes for the localized treatment of both early and advanced liver cancer.

However, this system has poor solution stability and low drug-loading efficiency. Microsol electrospinning can be used to achieve the efficient loading and slow release of hydrophilic drugs or easily deactivated biomolecules. Xia et al. successfully incorporated gemcitabine (GEM) into electrospun fibers using microsol electrospinning technology, resulting in a stable core–shell fibrous structure [135]. The release rate of GEM was modulated by adjusting the thickness of the hyaluronan-sol inner fiber and the quantity of the loaded GEM, leading to a sustained release for up to three weeks. Overall, this study presents a promising core–shell electrospun fiber approach for the controlled and localized delivery of GEM, which can enhance the treatment of residual tumors and prevent pancreatic tumor recurrence.

### 4.5. Coaxial Electrospinning

The coaxial electrospinning method can be used to encapsulate unstable bioactive molecules in electrospun fibers [81,136]. Compared to other electrospinning methods, coaxial electrospinning minimizes the interactions between organic polymer solutions and biomolecules, thus maintaining their biological activity. Coaxial electrospinning also allows for the simultaneous loading of hydrophilic and hydrophobic drugs, avoiding the biological toxicity caused by the late crosslinking of hydrophilic polymers. By loading different drugs or bioactive molecules into the core and sheath layers, coaxial electrospinning can fabricate a core–sheath structure that slows the release rate of the drug in the core layer [137].

Darbasizadeh et al. prepared polyethylene oxide/polycaprolactone core/sheath nanofibers via coaxial electrospinning and loaded them with doxorubicin hydrochloride (DOX) [138]. Sustained drug release and cytotoxicity against MCF-7 breast cancer cells were evaluated, and the nanofibers showed controlled release of the drug for up to 28 days, with no toxic effects on the cells. The DOX-loaded nanofibers exhibited high activity against MCF-7 breast cancer cells.

Rui et al. fabricated ultrathin C-SiC/ZrSixOyCz core–shell fibers using coaxial electrospinning and high-temperature pyrolysis with polycarbosilane as the core precursor and zirconium butoxide as the shell precursor [139]. The fibers exhibited excellent thermal stability and antioxidation properties owing to the protective effect of the shell layer and the self-healing mechanism. In addition, the fibers maintained good flexibility and deformability, indicating their potential as excellent thermal insulation materials for high-temperature applications.

### 4.6. Electrospray Technology

Electrospray technology can be used to integrate nanoparticles loaded with bioactive molecules or drugs onto or into fibers, allowing the deposition of particles to protect their activity and prolong the drug release. By combining electrospray and electrospinning technologies, spatiotemporally controlled drug release can be achieved, enabling the design of on-demand drug delivery systems that are responsive to external stimuli.

Xue et al. proposed a new approach to improve the mechanical properties of a fibrous electrospun nanofiber membrane by sandwiching it between two layers of electrospray-deposited microparticles [78]. The tensile strengths and elastic moduli of the composite membrane were significantly higher than those of the fibrous membrane. This study suggests that the layer-by-layer deposition technique using electrospraying and electrospinning has considerable potential in microfiltration applications.

Boda et al. introduced a new method for fabricating nanofiber microspheres using a combination of electrospinning and electrospraying techniques with potential applications in cell therapy [140]. This process involves electrospraying aqueous dispersions of electrospun nanofiber segments. The size and morphology of the microspheres can be controlled by varying the voltage applied during the electrospray process. This study demonstrated that nanofiber microspheres are better at carrying stem cells than solid microspheres and can form microtissue-like structures in situ when injected into microchannel devices.

Xue et al. used electrospray to form density gradients of biomacromolecular particles on nanofibers [141]. This method has been successfully applied to various types of biomacromolecules, including collagen, fibronectin, and laminin. For example, a gradient of collagen particles can promote the migration of bone marrow stem cells or NIH-3T3 fibroblasts in the direction of increasing particle density. This study suggests that scaffolds consisting of uniaxially aligned nanofibers and biomacromolecular particles in a density gradient can be used in a range of biological studies and biomedical applications.

In Table 4, the parameters and considerations for each drug-loading method are listed.

## 5. Biomedical Applications of the Therapeutic Agent-Loaded Fibrous Scaffolds

### 5.1. Scaffolds for Cardiovascular Regeneration and Vascularization

The cardiovascular system comprises the heart, blood vessels, and blood, which work together to circulate essential nutrients throughout the body and remove waste products. The heart is responsible for pumping blood throughout the body, while the blood vessels transport blood vessels transport blood, oxygen, nutrients, and other signaling molecules, including hormones. The cardiovascular system is involved in a wide range of functions, including the regulation of body temperature and maintenance of homeostasis. Therefore, it is necessary to regenerate the cardiovascular system properly when damaged. Injury to cardiac vascular tissue is generally accompanied by an insufficient blood supply to important tissues, causing health problems that could lead to death.

An example of a use for fibrous scaffolds for cardiovascular regeneration is in heart failure. Together with surgery, one strategy to cure infarcted hearts is the implantation of scaffolds for heart regeneration. Although the administration of cells that regenerate tissue could also be used for regeneration, the poor survival of the administered cells makes it difficult to induce the expected therapeutic effect [142]. Therefore, there is a persistent need to develop scaffolds to repair heart failure [143].

Richard et al. recently investigated therapeutic agent-coated PCL-based nanofiber-woven scaffolds for heart regeneration [144] (Figure 10). The woven scaffold, manufactured by electrospinning with a collecting device to yarn the fibers followed by weaving [145], exhibited suitable mechanical strength and could serve as an appropriate structure for cell adhesion. In addition, α-mangostin, a dip-coated woven scaffold, exhibited antioxidant scavenging activity that could promote cardiomyocyte growth and enhance heart repair [145,146,147,148]. In addition, the antioxidant activity on the surface could serve as a cardioprotective activity, resulting in the protection of myocardial tissues from oxidative stress. This synergistic effect between the fibrous scaffold structure and the therapeutic agent improved vascular generation and represented considerable potential for cardiac tissue regeneration.

In addition to implanting regenerating scaffolds, developing regeneration strategies to prevent unwanted platelet aggregation and thrombosis is crucial [149]. Thrombosis is blood clot formation that occurs in response to vascular injury, thereby preventing excessive bleeding. However, inappropriate thrombosis blocks blood vessels such as arteries that reach important organs, thereby inhibiting blood supply to the organs and causing serious health problems. Therefore, the development and functionalization of scaffolds that inhibit thrombosis and platelet adhesion to the scaffold surface should be considered [149,150].

With regard to this point, M. Guo et al. recently developed antithrombosis agent-loaded fibrous scaffolds for vascular tissue engineering [151] (Figure 11). Heparin, an antithrombosis agent used for clinical anticoagulant and antiplatelet activity in medical devices [152,153,154,155], was encapsulated in PCL fibers using the drug-mix electrospinning method. In addition, the fibers crystallize to construct a shish-kebab structure [156,157], which is expected to increase vascular endothelial cell adhesion and promote cell growth. The combination of heparin and shish-kebab structures can induce a synergistic effect in the generation of vascular tissue.

In the case of cardiac tissue regeneration, including the previously introduced research cases, most of the cardiac regeneration strategies are accompanied by new vascularization or angiogenesis [10,158,159,160]. Vascularization and angiogenesis often play a crucial role for cardiac regeneration as well as other tissue regeneration [161,162]. Vascularization is particularly important for larger tissues or organs to provide all the necessary nutrients and oxygen through blood vessels. Without the formation of proper blood vessels around the tissue, the engineered tissue cannot survive and maintain itself properly because of an inadequate supply of nutrients or oxygen. Therefore, to facilitate tissue regeneration, vascularization strategies are frequently combined with the construction of scaffolds that support the target tissue. In the following subsections, which discuss scaffolds for regenerating other tissues, such as nerve tissue, bone, or wounds, many developed scaffolds accompanying the induction of vascularization or angiogenesis for tissue regeneration are introduced.

### 5.2. Regenerating Agent-Loaded Fibrous Scaffolds for Nerve Regeneration

The nervous system consists of the central nervous system (CNS) and peripheral nervous system (PNS) and is the most important part of the human body that controls human actions and sensory information by communicating with different parts of the body using electrical signals or hormones [163]. Therefore, nervous system injuries can be a serious problem. For example, damage to the nerve tissue in the spinal cord can result in paralysis or loss of sensation in the limbs, whereas damage to the nerve tissue in the brain can result in cognitive or motor deficits [164].

It is necessary to regenerate damaged nerve tissue. However, nerve tissue has a limited capacity for self-repair [165]. Neurons have a limited ability to divide and regenerate. Therefore, developing an optimal environment for growth factor delivery and modulation is crucial for successful regeneration. Therefore, various types of therapeutic agent-delivering scaffolds have been developed.

Zhang et al. developed microRNAs (miRs) to deliver fibrous scaffolds for CNS and PNS regeneration [166]. MiRs are regulatory RNAs that modulate protein expression by RNA interference and can be used for neuroregeneration by manipulating local cell populations [167,168]. To prepare these fibrous scaffolds, they first prepared the polymeric fibrous scaffold made of PCL using the electrospinning method. The miRs were incorporated into the fiber after coating with polydopamine. The developed fibers can provide orientation and promote the growth of neurites by delivering miRs. When the developed fiber scaffold oriented in the collagen matrix was implanted in addition to transection in a spinal cord injury (SCI) model, nerve growth was observed in 2 weeks. In addition, the direction of neurofilament growth was parallel to the spinal cord by providing guidance in the direction owing to the aligned fibers.

In addition to nerve tissue treatment via gene delivery, small drug-loaded fibrous scaffolds can regenerate the nervous system. For example, deferoxamine (DFO)-loaded nerve scaffolds have the potential for peripheral nerve regeneration through the synergistic combination of drugs and scaffold orientation [169]. DFO, an iron chelator that stabilizes HIF1-α which enhances angiogenesis and suppresses radical oxygen species (ROS) such as hydroxyl radical, could be used for neuron regeneration by promoting angiogenesis with anti-inflammatory activity [170,171]. In addition, the scaffold construct of DFO-loaded fibers oriented in cylindrical PTFE rods served as nerve guidance conduits that guide axonal regrowth to induce nerve regeneration [169,172]. The combination of the orientation structure of the nanofiber and the DFO-releasing property could effectively enhance cell migration, vascularization, and accelerate axon growth (Figure 12). In this study, the orientation of the nanofiber could reduce the reduction of the classical M1 activation of the macrophages, whereas the oriented fiber structure could express proinflammatory factor TNF-α 0.7-fold less than the randomly structured fiber construct [169]. In terms of the oriented structure, the release of DFO activated tube formations in HUVECs resulted in rapid angiogenesis to promote nerve regeneration.

In addition to directly affecting the neurological system, immune regulation can regenerate the nerve system [173]. The classical M1 activation of macrophages could serve as a killing function for pathogens such as bacteria or tumor cells, accompanied by inflammation and downregulation of the proliferation of the surrounding tissue [174,175]. However, alternative M2 activation of macrophages could reduce the inflammation and induce the proliferation of the surrounding tissue [174,175]. Therefore, regulating the state of macrophages between M1 and M2 is a key factor for tissue regeneration, and this strategy could be used for nerve regeneration [176].

Consequently, Xi et al. prepared a macrophage-regulating fibrous scaffold for the regeneration of the nervous system [176] (Figure 13). The plasmid DNA (pDNA) encoding the DNA sequence to activate the macrophages toward the M2 phase was loaded into an aldehyde-containing cationic liposome via electrostatic interactions. Simultaneously, an amine-functionalized fibrous scaffold was manufactured using microsol electrospinning. The pDNA-loaded liposome was then loaded onto the surface of the scaffold by a condensation reaction between the amine of the scaffold and the aldehyde of the liposome. This pDNA-containing scaffold could regulate the M2-type polarization of macrophages and promote the neural differentiation of mesenchymal stem cells. In addition, when this scaffold was implanted into an SCI model, a significant reduction in the inflammatory response accompanied by a positive biological function of the nervous system regeneration, such as angiogenesis, which supplies nutrients and highly differentiated neurons, was observed within 8 weeks.

### 5.3. Bone Regeneration

Bone, a mineralized connective tissue, performs important functions such as mechanical support and shielding of important tissues in the body [177,178]. In addition to these functions, the bone marrow forms new blood cells, such as red blood cells, white blood cells, and platelets, to maintain the cardiovascular system. Therefore, the administration of proper regeneration strategies is necessary when bone defects occur.

Scaffold implantation into bone tissue is a common strategy used in bone tissue engineering and regeneration. The structure of the scaffold can be designed to mimic the natural structure of bone tissue, providing mechanical support, promoting osteoblast cell attachment and proliferation, and enhancing bone regeneration [179,180,181]. In addition to structural support, scaffolds can be loaded with various therapeutic agents to promote osteoblast differentiation, new vascular formation, or immune modulation, resulting in increased bone tissue regeneration. Therefore, in recent years, many strategies have been investigated for developing scaffolds loaded with therapeutic molecules for bone regeneration.

Geng et al. described the development of 3D scaffolds to enhance bone tissue growth by increasing vascularization (Figure 14). For the construction of the scaffold, poly(glycerol-cosebacic acid-co-L-lactic acid-co-polyethylene glycol) was mixed with salt particles to prepare a 3D-printing bioink. The bioink was printed using a 3D printer, and salt particles were removed after crosslinking. In the 3D-printed scaffold structure, macroporous gelatin nanofiber was accompanied by using the thermally induced phase separation (TIPS) technique [182]. After that, DFO, which induces angiogenesis, was loaded into a scaffold for bone regeneration. The developed scaffold exhibited a synergistic effect between the therapeutic activity of the drug and the skeletal support of the scaffold for bone regeneration. DFO increased the proliferation and migration of venous endothelial cells, thereby inducing vascularization surrounding the implanted tissue. Together with vascularization, the structural support of the scaffold could simultaneously promote osteogenesis and angiogenesis, thereby facilitating successful regeneration of the bone tissue.

In addition to scaffold manufacturing using 3D printing, electrospinning has been used for the development of bone regeneration scaffolds. Cui et al. recently developed DFO- and dexamethasone (DEX)-loaded fibrous scaffolds by coaxial electrospinning [15] (Figure 15). Coaxial electrospinning formed core–sheath layered fibers from different materials to sequentially release dual drugs [4,183]. DFO, an angiogenic agent, was rapidly released from the electrospun fibrous scaffold, which promoted vascularization surrounding the bone tissue to supply nutrients and oxygen. DEX, which modulates osteogenic differentiation and osteogenesis [184,185], was released sustainably during bone regeneration to differentiate bone mesenchymal stem cells into osteoblasts for a long time. This study could provide a potential strategy for the dual modulation of angiogenesis and osteogenesis to promote vascularized bone regeneration.

A promising approach to bone tissue engineering involves the use of small molecules to load drugs onto scaffolds and gene delivery systems to enhance the regeneration process. This has been demonstrated in recent studies and has the potential for effective and efficient bone repair treatments. Fang et al. developed plasmid DNA (pDNA)-loaded core–sheath-structured fiber scaffolds for bone defect repair [186] (Figure 16). The pDNAs corresponding to these sequences express two important growth factors, VEGF and bone morphogenetic protein 2 (BMP2). VEGF can induce vascularization around the tissue. BMP2 promotes bone tissue growth by stimulating osteogenic differentiation [187]. Together, these effects can lead to more effective and efficient bone regeneration, making this approach a promising avenue for future tissue-engineering research. For successful delivery of pDNA into the surrounding tissue, pDNAs were loaded onto the non-viral vector, and the vectors were then incorporated onto the electrospun PCL fiber sheath and alginate hydrogel core. This core–sheath structure of the scaffold could load two different therapeutic agents and release them sequentially, promoting osteogenesis and inducing successful vascularized bone regeneration.

In addition to applying vascularization and osteogenic differentiation to bone regeneration, osteoimmunomodulation of the surrounding bone tissue has the potential to promote bone regeneration. The activation of macrophages toward the inflammatory M1 phase is necessary during the initial stage of repair [188]. However, prolonged M1 phase activation can induce chronic inflammation, thereby damaging tissues. Therefore, the modulation of macrophages toward the anti-inflammatory M2 phenotype at the proper stage could be a promising strategy for successful bone regeneration [189]. Based on this strategy, He et al. developed an immunomodulatory nanomedicine-loaded fibrous membrane for enhanced bone regeneration [190] (Figure 17). The anti-inflammatory agents tannic acid (TA) and indomethacin (IND) [191,192] were assembled into a nanomedicine and released from a PCL-based nanofibrous scaffold after implantation. The release of TA and IND could modulate the macrophage phase toward the M2 phenotype and suppress inflammation, thereby exhibiting enhanced and effective bone regeneration.

### 5.4. Scaffolds for Wound Healing

A wound is an injury or break in the skin or underlying tissue caused by an external force, such as a cut, scrape, or burn [193]. Wounds can be acute or chronic and range in severity from minor cuts to deep lacerations, surgical incisions, or severe burns. In general, matrix components and other biological factors are activated to heal wounds [193,194]. However, pathological states such as severe burns or bacterial infections can induce a deficiency in wound healing [195]. In particular, severe burns or injuries cause defects in the skin layer, resulting in the infiltration of the pathogen deep inside the skin. Bacterial infections cause severe inflammation, delay tissue regeneration, and increase the risk of scarring.

Severe burns or bacterial infections can inhibit wound healing, and a patient’s chronic disease such as diabetes may also interfere with wound healing. Hyperglycemia resulting from diabetes can induce atherosclerosis, preventing the circulation of nutrients and impairing healing [196]. In addition, diabetes can weaken the immune system and cause a dysfunction of the immune response [197], resulting in increased bacterial infection and chronic inflammation, which inhibit wound healing. Therefore, scaffolds have been developed to promote wound healing.

Kanhassamy et al. reported that fibrous scaffolds containing antibacterial agents have the potential to promote effective wound healing by inhibiting additional bacterial infection [198] (Figure 18). Fibrous scaffolds produced by electrospinning that emulate the ECM can offer structural and biological support to expedite wound healing [199]. Carnosine peptides conjugated with vitamin K3 (VKC) exhibit antibacterial properties against both Gram-negative (e.g., *E. coli* and *Pseudomonas aeruginosa*) and Gram-positive (e.g., *Staphylococcus aureus*) bacteria. This agent promotes wound healing by inhibiting bacterial infections. Additionally, the VKC peptide can impart a hydrophilic surface to the scaffold, which can enhance cell attachment and differentiation, resulting in the faster vascularization of the neighboring tissue and wound-healing process [200,201]. To this end, the authors developed an electrospun fibrous scaffold consisting of SF and the VKC peptide. When this scaffold was implanted into a STZ-induced diabetic mouse model [202], it prevented bacterial infection and accelerated wound closure. This approach holds the potential for the development of advanced scaffolds for patients with diabetic wounds.

Similar to other biomedical applications, such as nerve and bone tissue regeneration, the modulation of the immune response could help drive the wound-healing process. When a wound occurs, inflammation is induced to protect the body from infection by microorganisms [203]. However, prolonged or excess inflammation can dysregulate matrix deposition and wound remodeling, resulting in hypertrophic scar formation, impaired wound regeneration, and chronic skin wounds [204,205,206]. Therefore, immune modulation is a promising strategy for successful wound healing. Among immune-regulating factors, interleukin-10 (IL-10) plays an essential role in decreasing the expression of inflammatory cytokines, regulating the ECM through fibroblast modulation, and promoting proper wound regeneration with normal physiological integrity [207,208,209]. Thus, IL-10 is a promising therapeutic agent for wound regeneration.

Due to the promising activity of IL-10, Chen et al. developed IL-10-incorporated electrospun fibrous scaffolds for complete wound regeneration [210] (Figure 19). The scaffolds were prepared by a microsol electrospinning method using recombinant IL-10 containing HA to encapsulate IL-10 in the core of the fiber. Along with electrospinning fabrication, additional IL-10 was immobilized on the fibers by plasma treatment [211], resulting in an initial burst release of IL-10 when the scaffolds were implanted in the wound. This core–shell structure releases IL-10 from the scaffold during the initial and subsequent phases of wound repair, resulting in scarless skin regeneration [210].

In addition to the research mentioned earlier, studies are being conducted on the utilization of scaffolds containing signaling molecules to facilitate tissue regeneration within the body. This approach has considerable potential for enhancing the efficacy of tissue regeneration and has generated considerable interest. For example, nitric oxide is a gaseous signaling molecule that promotes angiogenesis and antimicrobial activity depending on its concentration [212,213]. Therefore, it is expected that a nitric oxide-generating scaffold can be used for wound healing by inhibiting microbial infection.

Li et al. developed antimicrobial fibrous dressings for wound healing by using nitric oxide [214] (Figure 20). The wound-healing scaffold was prepared as follows. First, fibers containing PCL and gelatin were prepared via electrospinning. The fibers were then crosslinked in a genipin-containing solution. Genipin can interact with primary amines in gelatin, resulting in the formation of genipin-crosslinked gelatin scaffolds [215,216]. After the preparation of the scaffold, nitric oxide was charged into the scaffold to functionalize the N-diazenium diolates, which are nitric oxide-releasing moieties that have the potential for antibacterial activity and wound healing [217,218,219]. The resultant dressing scaffolds exhibited effective antimicrobial activities against *E. coli* and *Staphylococcus aureus* (*S. aureus*) with minimal cytotoxic effects against keratinocytes and fibroblasts. Although the wound-healing and antibacterial properties of the dressing scaffolds could be elucidated more clearly in vivo, these experimental results are expected to be applicable for wound dressings in the future.

### 5.5. Anticancer Drug-Loaded Fibrous Scaffolds for Inhibiting the Recurrence and Metastasis of Cancer

In the previous sections, we discussed various types of scaffolds for regenerating target tissues. Although the development of scaffolds is primarily focused on tissue regeneration, they can also be employed as adjunctive therapies to treat diseases. In this case, scaffolds can be implanted around the diseased tissue to enhance the efficacy of the treatment and inhibit disease recurrence.

Most therapeutic scaffolds have been developed for cancer treatments. Cancer, the second leading cause of death in the United States, is a major health problem worldwide [220]. Several methods such as surgery, chemotherapy, and radiotherapy have been used to treat cancer [221]. Among the treatment methods, surgery is the primary treatment for as much removal of the solidified tumor tissue as possible [222]. However, tumor metastasis and cancer recurrence remain serious problems in the successful treatment of cancer [223]. Therefore, the implantation of a fibrous scaffold that releases anticancer agents with surgery is a promising adjunctive strategy to treat cancer successfully [224].

Fang et al. developed a multiple-layered scaffold, including a combretastin A4 (CA4)-loaded 3D-printed scaffold and DOX-loaded electrospun fiber, for inhibiting tumor recurrence by the sequential release of the drugs [225] (Figure 21). The outer layer, the 3D-printed scaffold, induces hemolysis to inhibit bleeding after tumor tissue surgery. In addition, the release of CA4 from the scaffold blocks blood flow into the center of the tumor tissue, resulting in the inhibition of tumor growth. The DOX-loaded fiber, the inner layer of the scaffold, releases DOX to inhibit peripheral and metastatic tumor cells to inhibit cancer recurrence. The synergistic effect of bleeding, blocking blood flow, and the effect of the anticancer drugs exhibited an enhanced antitumor effect compared with other control groups that lack scaffolds or drugs.

In addition to the anticancer effect, the inhibition of bacterial infection is required for implanted scaffolds. Pathogen infection can have harmful effects on patients, resulting in decreased survival rates and increased healthcare cost [226]. Moreover, certain bacterial species reduce the anticancer effects of drugs [227]. For example, *Gammaproteobacteria*, present in pancreatic cancer tissue, can metabolize and inactivate GEM which inhibits tumor growth, resulting in a reduced efficacy of the anticancer effect [227]. Therefore, the incorporation of antitumor and antibacterial effects is a promising strategy to inhibit the tumor growth effectively.

Zhang et al. developed a fiber scaffold that included an anticancer agent and an antibacterial agent for the prevention of cancer recurrence [228] (Figure 22). To fabricate this scaffold, an anticancer agent GEM-loaded fibrous membrane was first prepared by electrospinning by blending GEM with poly-L-lactic acid (PLLA). Subsequently, the GEM-loaded membranes (GEM@PLLA) were immersed in a TA solution for TA coating on the membrane. Finally, silver nanoparticles (AgNPs) were incorporated onto the TA-coated GEM@PLLA by a TA-mediated dipping process. AgNPs are antibacterial agents that combat both Gram-positive and Gram-negative bacteria [229,230]. The AgNP-incorporated fibrous membranes (GEM@PLLA-TA-Ag) exhibited high antirecurrence effects on the cancer when implanted after tumor resection. In addition, the inhibition of tumor growth was considerably greater than that in the GEM-treated group in vivo.

### 5.6. Additional Therapeutic Applications of the Fibrous Scaffolds

In the previous sections, we have discussed the development of numerous fibrous scaffolds for various biomedical applications such as cardiovascular regeneration, nerve regeneration, bone regeneration, and wound healing, in addition to the use of fibrous scaffolds to inhibit disease recurrence. In addition, fibrous scaffolds can also be used for the regeneration of other tissues.

Another biomedical application is periodontal regeneration [231]. Periodontal tissue (periodontium), a specialized oral tissue that plays a crucial role in providing support to the teeth, is composed of a sophisticated structure comprising hard tissues, such as bone, and soft tissues, such as gingiva [232]. Therefore, the strategies for regenerating periodontal tissue resemble those for regenerating bone [231,232]. Osteogenesis and vascularization have been used to regenerate the alveolar bone that supports the teeth [231,233]. In addition to bone regeneration, the wound-healing process is also applicable for regenerating the soft tissue that covers the periodontium [2,233,234].

Another possible application of fibrous scaffolds is in tendon regeneration [1,235,236,237]. The tendon, a connective tissue between the muscle and bone, plays a crucial role in the musculoskeletal system [235]. Globally, over 15 million people suffer from tendon injuries annually, and numerous therapeutic strategies have been suggested for tendon repair [235,236]. However, successful tendon repair is yet to be achieved [1,235]. Therefore, the development of scaffolds for tendon regeneration has been researched [1,237,238]. In a recent study, fibrous scaffolds loaded with angiogenic agents were shown to increase vascular formation near the implanted tendon site, promoting tendon regeneration [237].

In addition to the aforementioned cases, extensive studies have been conducted using fibrous scaffolds loaded with therapeutic agents for a wide range of tissue regeneration and disease treatment applications. To provide a comprehensive overview, we present our findings in Table 5.

## 6. Future Perspective and Limitations

The growing demand for the regeneration of injured or diseased tissues has led to the emergence of tissue engineering as a promising field. Tissue engineering uses scaffolds to provide structural and biological support for tissue regeneration [8]. The mechanical support of the scaffold facilitates cell attachment and proliferation, resulting in tissue regeneration with a proper orientation [258,259]. The scaffold can also be utilized as a reservoir for loading and releasing bioactive agents, such as drugs, genes, proteins, and peptides, leading to efficient and effective tissue regeneration. Therefore, the development of therapeutic agent-loaded scaffolds is a promising strategy for achieving successful tissue regeneration.

Despite their potential, fibrous scaffolds loaded with therapeutic agents have not yet been approved for primary use. To fully realize their clinical application, various factors, such as the potential toxicity of solvents, crosslinkers, or materials must be thoroughly assessed [211,260]. Regulatory guidelines should also be considered to develop a clinically usable scaffold in the future [211,261]. Although pre-clinical studies have been conducted to clarify the therapeutic effects of fibrous scaffolds, the significant differences between humans and animals tested in these studies could lead to unexpected results during clinical trials [261,262]. One example is the notable variations in the immune systems of mice and humans. This difference implies that the immune responses observed during pre-clinical tests in mice may not correspond to those observed in humans [261,263]. Therefore, it is necessary to prepare humanized animal models to evaluate the therapeutic effects and biocompatibility of scaffolds for further clinical trials [264,265]. Along with the regulatory aspects and proper strategies to reveal pre-clinical data, mass production with the reproducibility of the scaffold must also be considered for clinical applications and market release [211,262]. One example of a commercialized fibrous scaffold is Biogide, a collagen membrane used for guided tissue regeneration in dentistry. One limitation of commercialized fibrous scaffold is the lack of optimal mechanical properties and biocompatibility for some specific applications. Additionally, the cost of production and regulatory approval can be significant barriers to commercialization. Furthermore, the degradation rate and stability of the scaffold material may not be suitable for long-term applications. These factors can limit the widespread use of commercialized fibrous scaffolds in regenerative medicine.

Despite existing challenges, therapeutic agent-loaded fibrous scaffolds hold immense promise for future clinical applications. The future direction of regenerative fibrous scaffolds may involve addressing the limitations of current scaffolds and exploring new approaches to scaffold design and fabrication. This may include developing novel biomaterials with improved mechanical properties, biocompatibility, and degradation rates, as well as incorporating advanced technologies such as 3D printing and microfluidics to achieve more precise control over scaffold structure and function. In addition, by addressing various factors such as regulatory guidelines, mass production, reproducibility, and animal studies that accurately mimic human conditions, the development of advanced scaffolds with superior tissue regeneration capabilities is expected to be successful.

## 7. Conclusions

In this review, we present an overview of the use of fibrous scaffolds loaded with therapeutic agents in tissue engineering. Various methods can be used to prepare fibrous scaffolds, including electrospinning methods and 3D printing. Electrospinning is the most commonly used method for preparing fibrous scaffolds because it allows for the facile processing of a wide range of materials into fibrous scaffolds. Three-dimensional printing technology has recently been used to fabricate fibrous scaffolds to create customized structures that closely match the target tissue. These scaffolds are loaded with a range of therapeutic agents to enhance their regenerative effects upon implantation. Various methods have been used to load therapeutic agents into scaffolds, such as pre-mixing before scaffold fabrication, physical adsorption or surface coating, and immobilization after scaffold fabrication.

Furthermore, this review focuses on the recent therapeutic applications of therapeutic agent-loaded fibrous scaffolds. These scaffolds have been utilized for various tissue regeneration applications such as cardiovascular tissue, nerves, bones, and wound healing. Strategies such as vascularization, growth factor stimulation, and immune modulation have been implemented to achieve effective tissue regeneration. Additionally, these scaffolds have been used to treat diseases such as tumors and microbial infections. Scaffolds can help prevent tumor recurrence and inhibit bacterial infections to enhance therapeutic efficacy and effectively treat tissues. The results of these studies demonstrate the diverse potential of therapeutic agent-loaded fibrous scaffolds and provide valuable insights into the development of advanced scaffolds that can be used for the treatment of other diseases or tissue regeneration in the future.

## Figures and Tables

**Figure 1 pharmaceutics-15-01522-f001:**
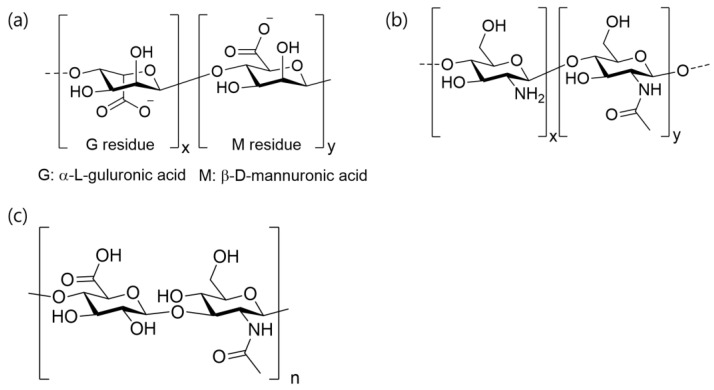
Chemical structures of polysaccharides. (**a**) Alginate [17,18], (**b**) chitosan [19,20], (**c**) hyaluronic acid [21].

**Figure 5 pharmaceutics-15-01522-f005:**
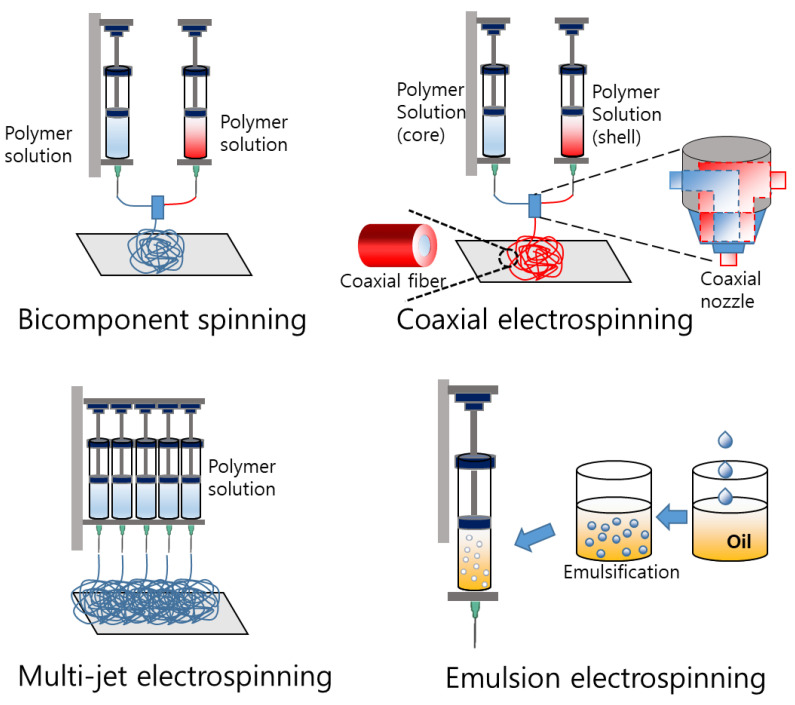
Schematic illustration of electrospinning methods.

**Figure 6 pharmaceutics-15-01522-f006:**
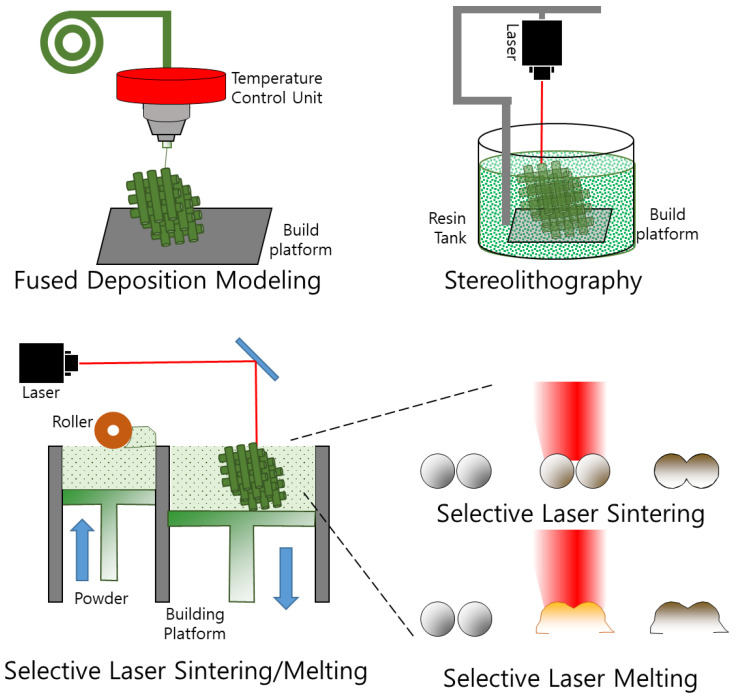
Schematic illustration of various 3D printing methods.

**Figure 7 pharmaceutics-15-01522-f007:**
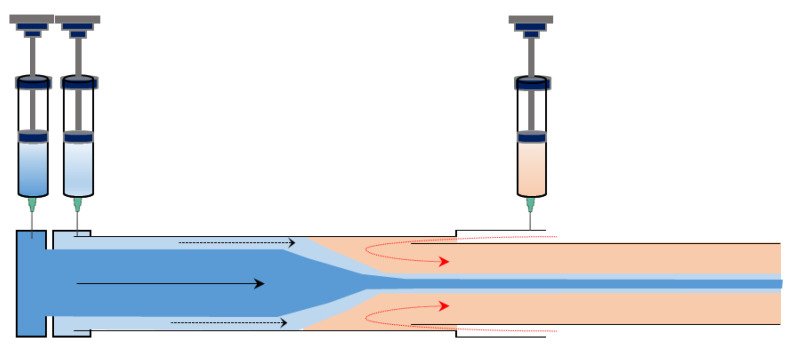
Example of fabrication of fiber with glass capillaries (Modified from ref. [100]. Copyright 2019 Royal Society of Chemistry).

**Figure 8 pharmaceutics-15-01522-f008:**
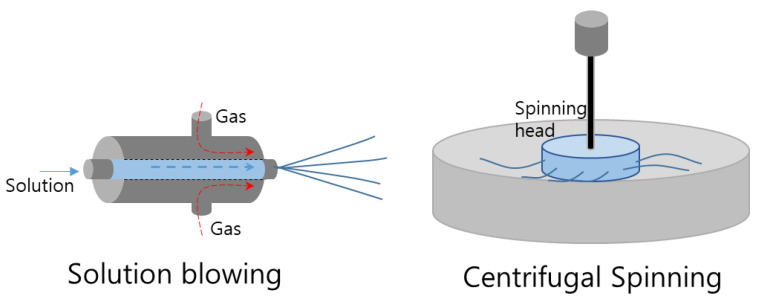
Schematic illustration of solution blowing and centrifugal spinning.

**Figure 9 pharmaceutics-15-01522-f009:**
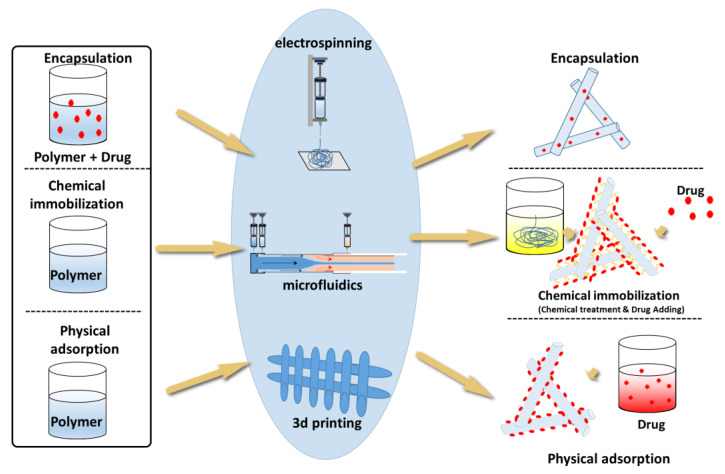
Schematic of various methods of drug loading.

**Figure 10 pharmaceutics-15-01522-f010:**
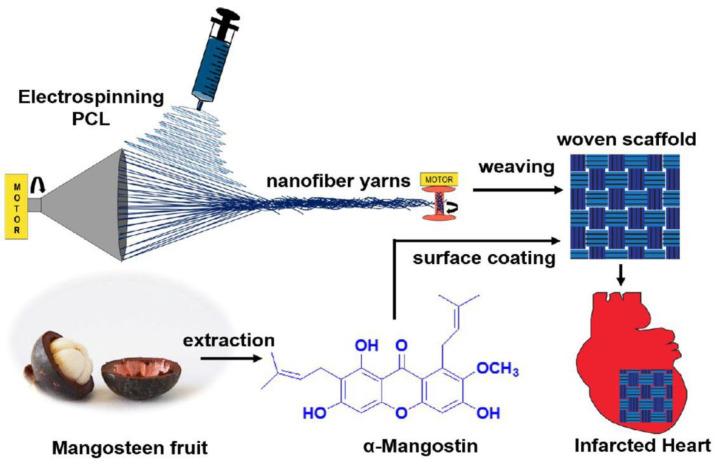
Schematic illustration of the antioxidant agent α-mangostin-coated fibrous woven scaffold for heart regeneration. (Reprinted with permission from ref. [144]. Copyright 2022 American Chemical Society).

**Figure 11 pharmaceutics-15-01522-f011:**
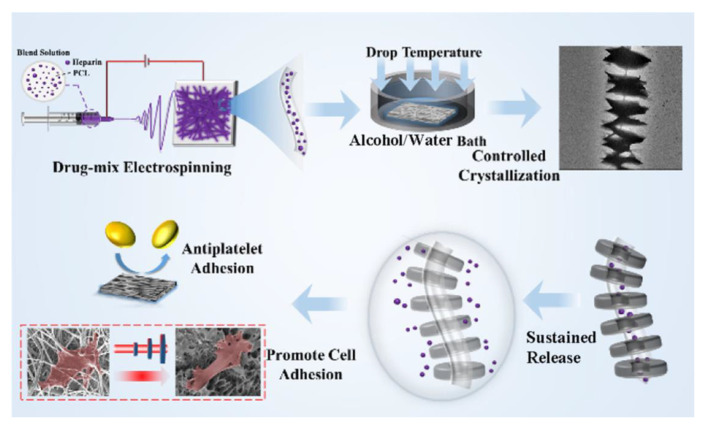
Overall scheme of the heparin encapsulated fiber-based scaffold for inhibiting thrombosis with enhanced vascular regeneration. (Reprinted with permission from ref. [151]. Copyright 2022 American Chemical Society).

**Figure 12 pharmaceutics-15-01522-f012:**
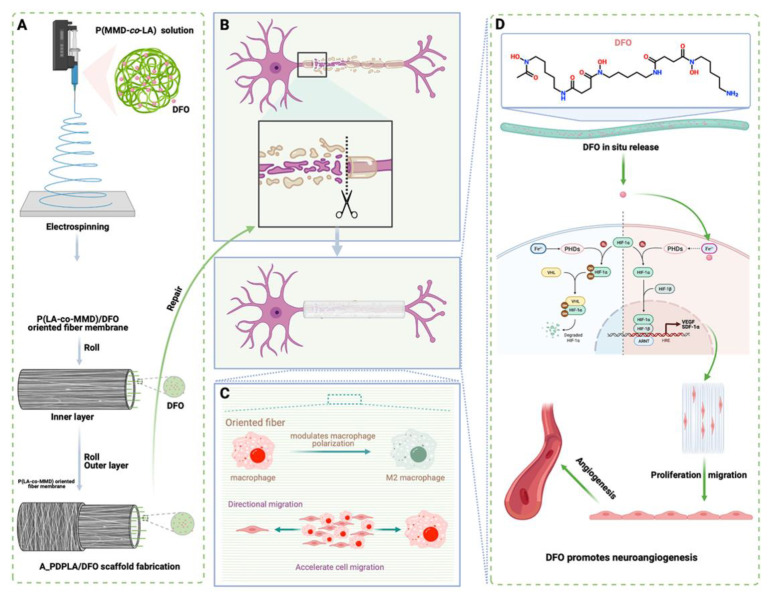
Overall scheme of the oriented DFO-loaded fibrous scaffold and its activity for regenerating nerve tissue. (Reprinted with permission from ref. [169]. Copyright 2021 Elsevier). (**A**) Manufacturing process of DFO loaded nerve scaffolds. (**B**) Schematic illustration of peripheral nerve injury and nerve regeneration after scaffold implantation. (**C**) Schematic illustration of functions such as modulation and migration in oriented fibers. (**D**) Mechanism of the vascularization by DFO.

**Figure 13 pharmaceutics-15-01522-f013:**
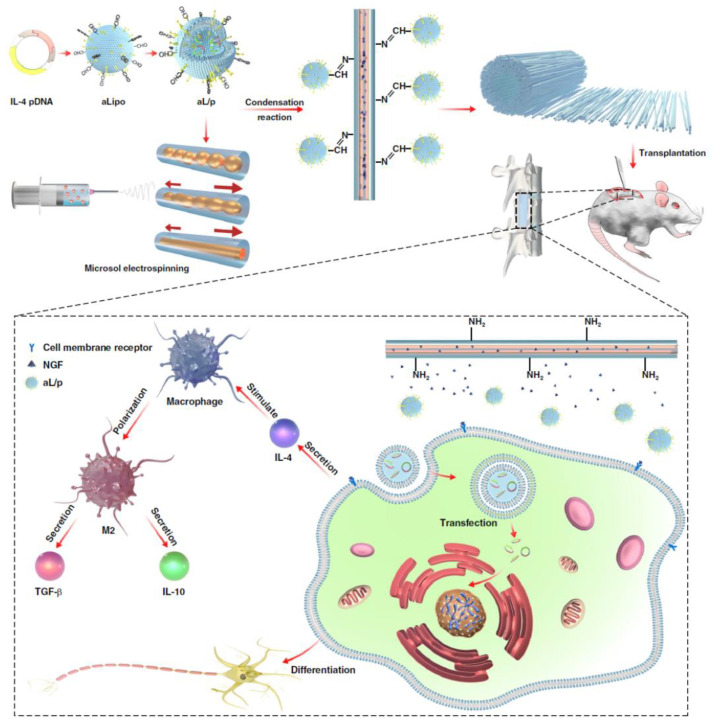
Fabrication scheme of fibrous scaffold accompanying pDNA-loaded nanoparticle and the mechanism for nerve regeneration. (Reprinted from ref. [176]. Copyright 2020 Springer Nature).

**Figure 14 pharmaceutics-15-01522-f014:**
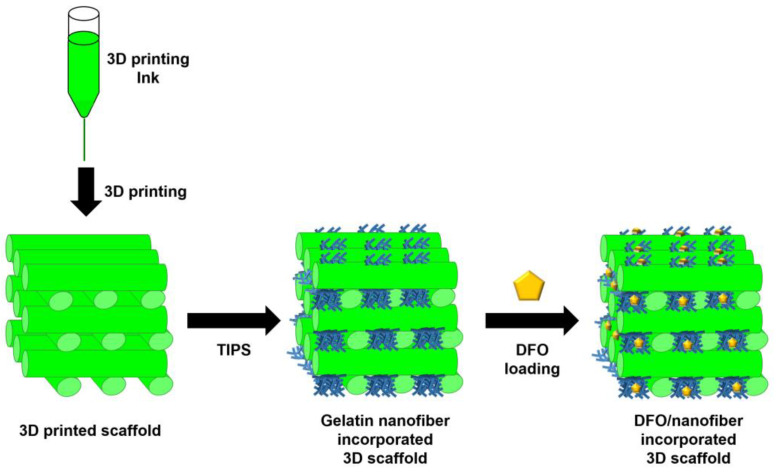
Schematic illustration of the 3D-printed scaffold manufacturing for bone tissue regeneration (Modified from ref. [182]. Copyright 2021 Royal Society of Chemistry).

**Figure 15 pharmaceutics-15-01522-f015:**
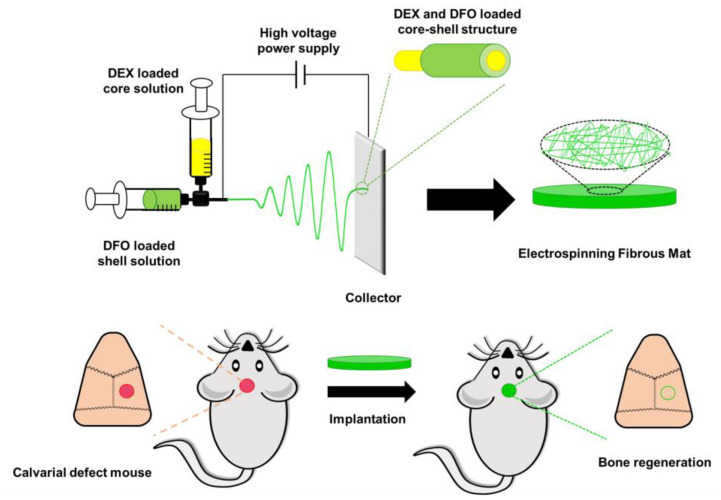
Schematic illustration of the DEX- and DFO-loaded fibrous scaffold for vascularized bone regeneration (Modified from ref. [15]. Copyright 2022 Wiley).

**Figure 16 pharmaceutics-15-01522-f016:**
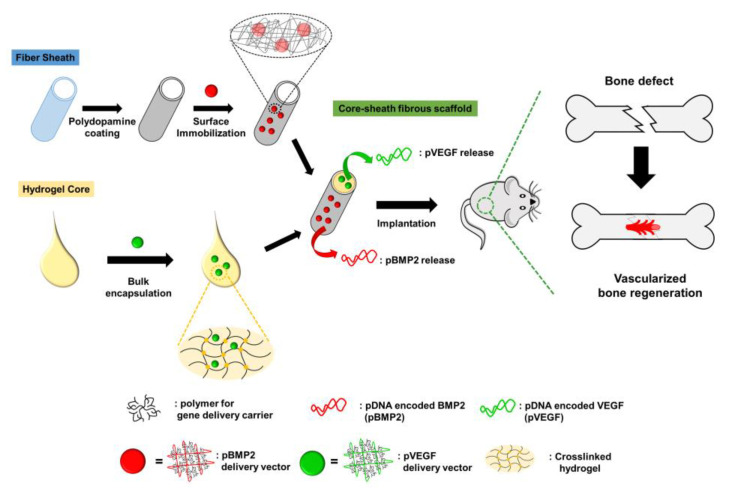
Schematic illustration of scaffold containing electrospun PCL fiber sheath and alginate hydrogel for vascularized bone regeneration. pDNAs that generate BMP2 and VEGF are loaded into the sheath and core, respectively, and the release of pDNAs could induce vascularization and osteogenesis for bone regeneration (Modified from ref. [186]. Copyright 2022 Wiley).

**Figure 17 pharmaceutics-15-01522-f017:**
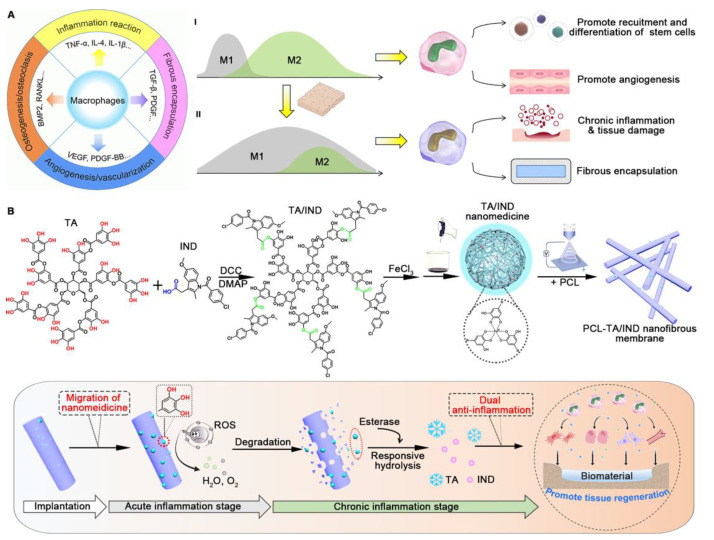
Schematic illustration of the osteoimmunomodulation of the fibrous scaffolds for bone regeneration. (Reprinted with permission from ref. [190]. Copyright 2022 American Chemical Society.) Macrophage phase to M2 macrophage in late stage suppresses inflammatory response, thereby displaying effective bone regeneration. The drugs, TA and IND, act as anti-inflammatory agents. (**A**) Schematic illustration of the macrophage = mediated immune response, demonstrating the normal healing process (I) and abnormal immune response such as prolonged M1 macrophage activation (**B**) Preparation method for TA and IND loaded nanofibrous scaffold and its immune-modulating function.

**Figure 18 pharmaceutics-15-01522-f018:**
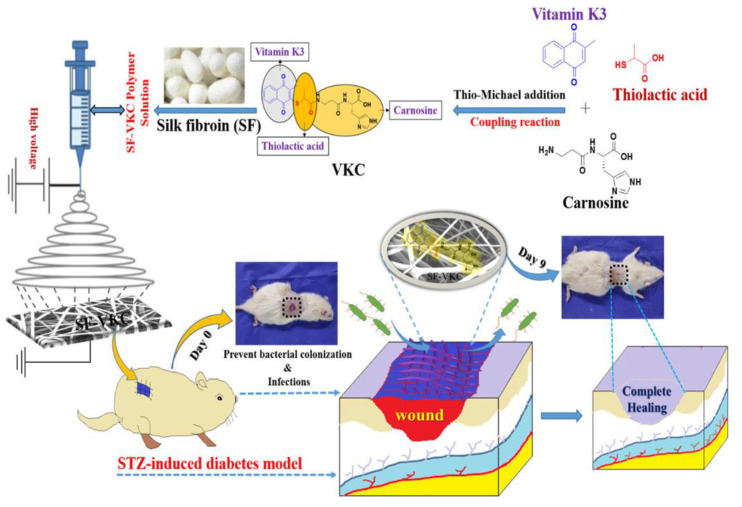
Synthetic scheme of the silk fibroin-based fibrous scaffold for wound healing. The drug VKC provides antibacterial activity for inhibiting additional bacterial infection. The fibrous scaffold serves as not only an antibacterial reservoir but also promotes wound-healing processes. (Reprinted with permission from ref. [198]. Copyright 2021 American Chemical Society).

**Figure 19 pharmaceutics-15-01522-f019:**
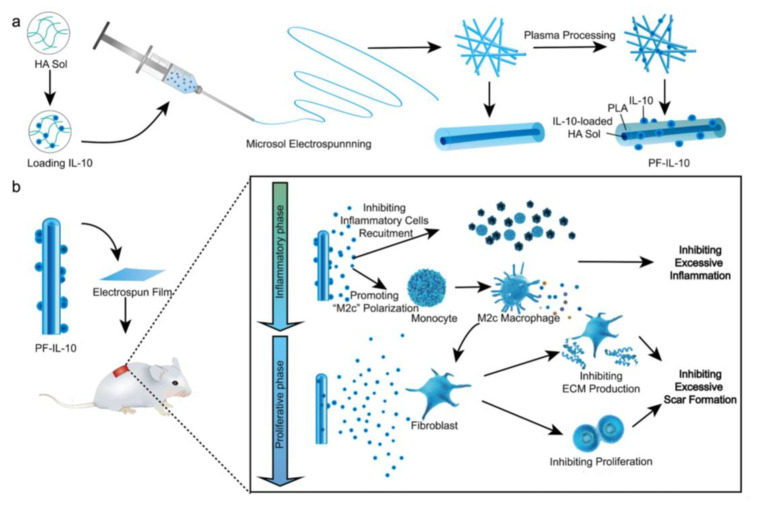
Schematic illustration of the immune-modulating electrospun scaffolds for wound healing. The protein IL-10 released from scaffolds could differentiate macrophages toward to M2 phages and activate fibroblasts, resulting in inactivating inflammation with reduced scar formation. (Reprinted with permission from ref. [210]. Copyright 2021 Elsevier) (**a**) Manufacturing process of IL-10 loaded scaffolds combined with electrospinning method and plasma treatment. (**b**) Schematic illustration of wound healing process of immune-modulating scaffolds.

**Figure 20 pharmaceutics-15-01522-f020:**
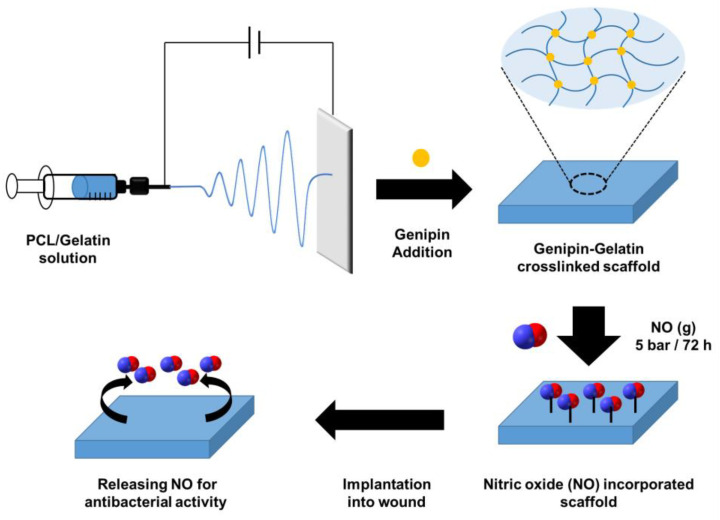
Fabrication of the nitric oxide-incorporated electrospun scaffolds for antibacterial infection (Modified from ref. [214]. Copyright 2022 ACS Publications).

**Figure 21 pharmaceutics-15-01522-f021:**
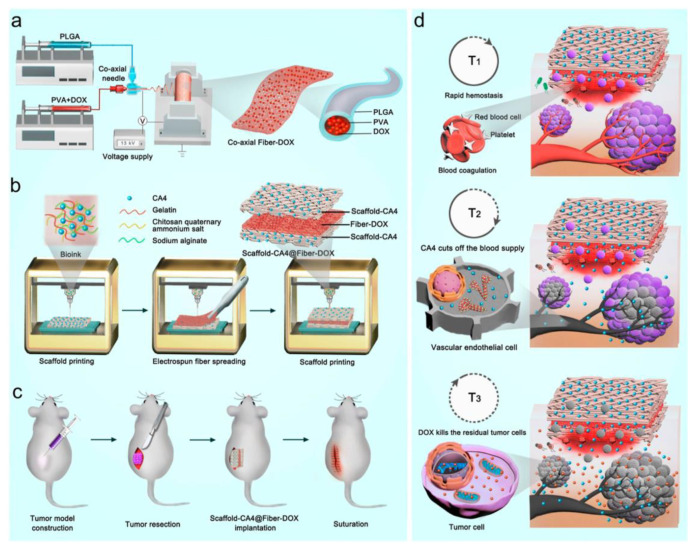
Overall scheme of the multiple-layered scaffold containing dual anticancer drugs for inhibiting tumor reoccurrence. (Reprinted with permission from ref. [225]. Copyright 2022 American Chemical Society) (**a**) Manufacturing process of Dox loaded co-axial fibers by electrospinning method. (**b**) Manufacturing process of sandwich like scaffold by 3D printing. (**c**) Schematic illustration of the tumor modeling and scaffold implantation. (**d**) Schematic illustration of the tumor therapy induced by CA4 and DOX.

**Figure 22 pharmaceutics-15-01522-f022:**
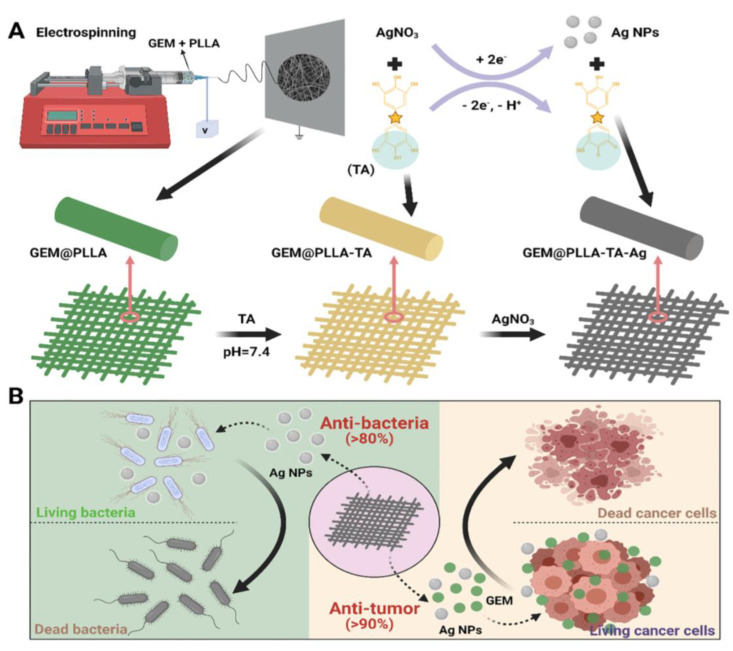
Schematic illustration of the electrospun fibrous membrane containing antibacterial agent with anticancer drugs for synergistic tumor inhibition treatment. (Reprinted with permission from ref. [228]. Copyright 2022 Elsevier) (**A**) Manufacturing process of AgNPs incorporated fibrous membrane (GEM@PLLA-TA-Ag). (**B**) Schematic illustration of anti-bacterial effect and anti-tumor effect of GEM@PLLA-TA-Ag.

**Table 1 pharmaceutics-15-01522-t001:** Comparison of electrospinning methods.

Spinning Method	Key Features	Applications
Bicomponent spinning [79,80]	Two polymers are spun together to create a core–sheath or side-by-side structure	Medical textiles, filtration, and protective clothing
Coaxial electrospinning [81,82,83]	Two or more fluids are spun together in a coaxial arrangement, producing fibers with a core–shell structure	Drug delivery, tissue engineering, and energy storage
Multijet electrospinning [83,84,85]	Multiple spinning nozzles are used simultaneously to produce a large volume of fibers in a short amount of time	Tissue engineering, drug delivery, and energy storage
Emulsion electrospinning [86,87,88]	An emulsion of two immiscible liquids is spun to create fibers with a polymer and a liquid core	Drug delivery, tissue engineering, and food packaging

**Table 2 pharmaceutics-15-01522-t002:** Comparison of methods of 3D printing.

Methods	Principles	Materials Used	Advantages	Disadvantages
Fused deposition modeling [89,90]	Thermoplastic filaments are melted and deposited layer by layer using a heated nozzle	ABS, PLA, Nylon, TPU, etc.	Low cost, user-friendly, wide material compatibility, good for prototyping	Lower resolution, visible layer lines, limited to certain geometries, weaker mechanical properties
Stereolithography (SLA) [91,92]	Uses a UV laser to selectively cure a liquid photopolymer resin layer by layer	Resin materials (acrylic, epoxy, polyurethane, etc.)	High resolution, good surface finish, suitable for small and intricate parts, good for prototyping and production	Limited material selection, expensive, post-processing required, not suitable for large parts
Selective laser sintering (SLS) [93,94,95]	Uses a laser to selectively fuse a powdered material (such as nylon) layer by layer	Nylon, TPU, polycarbonate, etc.	No support structures required, can produce complex geometries, good mechanical properties, good for low-volume production	Limited surface finish, expensive, post-processing required, not suitable for small parts
Selective laser melting [96]	Similar to SLS, but uses a laser to fully melt the powdered material instead of just fusing it, resulting in fully dense metal parts	Metals such as titanium, aluminum, steel, etc.	High strength, good for small and complex parts, wide range of materials available	Expensive, limited build size, slow printing speed, post-processing required

**Table 3 pharmaceutics-15-01522-t003:** Parameters and considerations for each fabrication method for fibrous scaffolds.

Fabrication Methods	Parameters	Considerations
Electrospinning [119,120,121]	Solution viscosity	Higher viscosity results in larger polymer beads and thicker fibers
	Solution conductivity	Higher conductivity improves the quality of fibers
	Applied voltage	Higher voltage results in thinner fibers
	Distance between the collector and the needle	Shorter distance results in denser fibers
	Flow rate of the solution	Higher flow rate leads to larger fiber diameter
	Needle diameter	Smaller diameter results in thinner fibers
	Solution concentration	Higher concentration results in thicker fibers
	Humidity and temperature of the environment	Affects fiber morphology and diameter
3D Printing [89,90,91,92,93,94,95,96]	Material properties	Affects printability and mechanical properties
	Printing speed	Affects print quality and resolution
	Layer height	Affects resolution and mechanical properties
	Temperature of the printing environment	Affects material properties and print quality
	Extruder speed	Affects flow rate and print quality
	Nozzle diameter	Affects resolution and print speed
	Printing bed temperature	Affects adhesion and warping
	Printing orientation	Affects mechanical properties
Microfluidic method [100,106,107,108,109,110,111]	Flow rate of the fluid	Affects fiber diameter and porosity
	Viscosity of the fluid	Affects flow rate and fiber morphology
	Channel geometry	Affects fiber diameter and alignment
	Material properties	Affects fiber morphology and mechanical properties
	Surface properties	Affects fiber adhesion
	Temperature and pressure of the system	Affects fiber morphology and diameter
Solution blowing [113,114,115]	Solution viscosity	Higher viscosity results in thicker fibers
	Gas flow rate	Affects fiber diameter and morphology
	Distance between the nozzle and the collector	Affects fiber alignment and density
	Solution concentration	Higher concentration results in thicker fibers
	Temperature and humidity of the environment	Affects fiber morphology and diameter
Centrifugal spinning [116,117,118]	Polymer concentration	Higher concentration results in thicker fibers
	Centrifugal force	Affects fiber diameter and alignment
	Distance between the collector and the nozzle	Affects fiber alignment and density
	Solution viscosity	Higher viscosity results in thicker fibers
	Solution flow rate	Affects fiber diameter and morphology
	Polymer molecular weight	Affects mechanical properties
	Temperature and humidity of the environment	Affects fiber morphology and diameter

**Table 4 pharmaceutics-15-01522-t004:** Parameters that should be considered for each drug-loading method.

Method	Parameters	Considerations
Blending [122,123,124,125,126,127]	Solubility of drug and polymer	Drug and polymer should have similar solubilities
	Drug and polymer concentration	Higher concentrations lead to higher drug loading
	Mixing time and speed	Affects drug–polymer interaction and drug distribution
Chemical immobilization [128,129]	Type of chemical reaction	Reaction should be specific to the drug and polymer
	Concentration of reactants	Higher concentrations lead to higher drug loading
	Reaction time and temperature	Affects the efficiency of the reaction and drug loading
Physical adsorption [130,131,132]	Surface area of the scaffold	Larger surface area leads to higher drug loading
	Solution concentration and pH	Affects drug–polymer interaction and adsorption efficiency
	Adsorption time and temperature	Affects the efficiency of drug adsorption
Emulsion electrospinning [133,134,135]	Polymer and drug solubility	Solubility should be matched for codissolving
	Emulsifier concentration	Affects droplet size and stability
	Electrospinning parameters	Affects fiber diameter and drug distribution
	Emulsion stability	Emulsion should be stable for efficient drug loading
Coaxial electrospinning [137,138,139]	Core and shell polymer solubility	Solubility should be matched for codissolving
	Core and shell polymer concentration	Higher concentrations lead to higher drug loading
	Coaxial electrospinning parameters	Affects fiber diameter, shell thickness, and drug distribution
	Core–shell fiber stability	Core and shell should remain stable during processing
Electrospray [140,141]	Solution concentration and pH	Affects drug–polymer interaction and encapsulation efficiency
	Flow rate and voltage	Affects droplet size and drug distribution
	Solvent type	Affects the solubility of the drug and polymer
	Post-processing conditions	Can affect drug loading and release kinetics

**Table 5 pharmaceutics-15-01522-t005:** Applications of fibrous scaffolds.

Biomedical Applications	Fabrication Methods of Scaffold	Therapeutic Agent (with its Function)	Drug-Loading Methods	References
Cardiovascular regeneration	Electrospinning method	α-Mangostin (antioxidant, cardioprotective activity)	Dip coating	[144]
Cardiovascular regeneration	Electrospinning method with crystallization	Heparin (antithrombosis activity)	Drug-mixing (blending) electrospinning method	[151]
Vascular regeneration	Coelectrospinning method	IL-4 (immune modulation for vascular stabilization)	Surface modification of the scaffold	[239]
Cardiac regeneration	Electrospinning method	Resveratrol (inducing cardioprotective effect)	Drug-mixing electrospinning method	[240]
Nerve regeneration	Electrospinning method	MicroRNAs (miRNAs, genetic modulation for nerve regeneration)	Passive loading after scaffold fabrication	[167]
Nerve regeneration	Electrospinning method with rolling	Deferoxamine (promoting angiogenesis with anti-inflammation)	Drug-mixing electrospinning method	[169]
Nerve regeneration	Microsol electrospinning	IL-4 encoding plasmid DNA (IL-4 generation for immune suppression)	Chemical modification of pDNA-loaded liposome	[176]
Nerve regeneration	Electrospinning method	Magnetic nanoparticles (MgO, inhibiting nerve cell apoptosis), purmorphamine and retinoic acid (Pur/RA, inducing neuronal differentiation)	Pur/RA-loaded MgO-mixing electrospinning method	[241]
Nerve regeneration	Electrospinning method	Acidic fibroblast growth factor (aFGF, signaling molecule for axon growth)	Coaxial electrospraying of aFGF-loaded nanoparticles	[242]
Nerve regeneration	Microsol electrospinning method	Brain-derived neurotrophic factor (BDNF, improving differentiation of bone marrow mesenchymal stem cells into neurons)	Drug-mixing electrospinning method	[243]
Bone regeneration	3D printing with TIPS technique	Deferoxamine (DFO, inducing angiogenesis for vascularization)	Passive drug loading after scaffold fabrication	[182]
Bone regeneration	Coaxial electrospinning method	DFO (inducing vascularization), dexamethasone (DEX, modulating osteogenic differentiation)	Drug-mixing electrospinning method	[15]
Bone regeneration	Electrospinning method	Vascular endothelial growth factor (VEGF) encoding pDNA (inducing vascularization), bone morphogenetic protein 2 (BMP2, promoting growth of bone tissue)	Surface immobilization of drug-loaded vector and embedding drug containing hydrogel inside the scaffold	[186]
Bone regeneration	Electrospinning method	Tannic acid (TA) and indomethacin (IND) (agents for anti-inflammation)	TA/IND nanomedicine-mixing electrospinning method	[190]
Bone regeneration	Coaxial electrospinning method	Tauroursodeoxycholic acid (TUDCA, inducing vascularization), BMP2 (promoting growth of bone tissue)	Drug-mixing electrospinning method	[244]
Bone regeneration	Electrospinning method	Recombinant human vein endothelial growth factor (rhVEGF, inducing vascularization), recombinant human bone morphogenetic protein 2 (rhBMP2), and calcium phosphates (inducing bone tissue regeneration)	Drug-mixing electrospinning method	[245]
Bone regeneration	Electrospinning method with crosslinking	IL-4 (immunomodulation for vascular maturation and osteogenesis)	Drug-mixing electrospinning method	[246]
Wound healing	Electrospinning method	Vitamin K3 carnosine peptide (agent for antibacterial activity)	Drug-mixing electrospinning method	[198]
Wound healing	Microsol electrospinning method	IL-10 (agents for anti-inflammatory effect)	Drug-mixing electrospinning method	[210]
Wound healing	Electrospinning method	Nitric oxide (NO, agent for antibacterial activity)	Surface modification after scaffold fabrication	[214]
Wound healing	Electrospinning method	*Cordia myxa* (agent for antibacterial activity with antioxidant effect)	Immobilization after scaffold fabrication	[247]
Wound healing	Electrospinning method	Anemoside B4 (ANE, agent for anti-inflammation)	Incorporation of ANE after scaffold fabrication	[248]
Wound healing	3D printing	Vascular endothelial growth factor (VEGF, inducing vascularization), gentamicin (antibiotic agent), silver nanoparticle (antibacterial agent)	Drug loading in 3D printing ink	[249]
Wound healing	Physical coassembling to construct fibrous hydrogel	Phycocyanin (anti-inflammatory activity), gallic acid (antibacterial activity and anti-inflammatory activity)	Mixing drugs for assembled fibrous scaffold fabrication	[250]
Wound healing	Electrospinning method	Manganese dioxide (MnO_2_, agent for assuaging oxidative stress), curcumin (agent for anti-inflammation)	Curcumin-loaded MnO_2_ nanoparticle-mixing electrospinning method	[251]
Wound healing	Electrospinning method	Pyrogallol and saikosaponin (reducing reactive oxygen species (ROS))	Drug-mixing electrospinning method	[252]
Wound healing	Electrospinning method	Curcumin (agent for anti-inflammation), cerium nitrate (reducing ROS for antiscar wound healing)	Drug-mixing electrospinning method	[253]
Wound healing	Layer by layer electrospinning method	Silver (I) sulfadiazine (agent for antibacterial activity)	Drug-mixing electrospinning method	[254]
Tumor inhibition	3D printing with electrospinning method for multiple-layered scaffold fabrication	Combretastin A4 (CA4, inhibiting tumor growth) and doxorubicin (DOX, inhibiting metastasis and recurrence of cancer)	DOX: drug-mixing electrospinning method, CA4: drug-mixing in bioink before 3D printing	[225]
Tumor inhibition with antibacterial activity	Electrospinning method	Gemicitabine (GEM, inhibits tumor growth), silver nanoparticle (antibacterial agent for enhancing tumor therapy)	GEM: drug mixing electrospinning method, silver nanoparticle: dipping process after scaffold fabrication with coating	[228]
Tumor inhibition	Sol–gel process with electrospinning method	Glucose oxidase (GOx, inducing starvation activity around tumor tissue), hyaluronidase (HAase, destroying tumor extracellular matrix (ECM) to penetrate nanomedicine), banoxantrone (AQ4N, antitumor agent)	Incorporation of HAase and GOx/AQ4N-loaded nanoparticles after scaffold fabrication	[255]
Tumor inhibition	Coaxial 3D printing	DOX (antitumor agent), polydopamine (photothermal agent for thermal-induced tumor therapy)	Mixing drugs in ink and fabricating scaffold	[256]
Tumor inhibition with wound healing	Coaxial electrospinning method	Fluorouracil (5-FU, agent for chemotherapy of melanoma)	5-FU-loaded nanoparticle-mixing electrospinning method	[257]
Periodontal regeneration	Electrospinning method	Dimethyloxalylglycine (DMOG, inhibitor of prolyl hydroxylases to promote VEGF)	Drug-mixing electrospinning method	[231]
Tendon regeneration	Electrospinning method	Nitric oxide (NO, agent for vascularization)	Use of NO-loaded metal–organic framework-mixing electrospinning method	[237]

## Data Availability

Not applicable.

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
