# Peer review of "Therapeutic Agent-Loaded Fibrous Scaffolds for Biomedical Applications"

_pharmaceutics, 2023, doi:10.3390/pharmaceutics15051522_

Round 1
Reviewer 1 Report
The aim of the work should be clearly stated. A review is usually part of a larger task. It is possible to create some new technology or improve the known ones.
".....Furthermore, collagen can be easily extracted from natural sources such as bovine hide, pig skin, and human placenta, making it a readily available and cost-effective material for scaffold production". row 170-172
The authors, referring to a single source, identified the human placenta as a cost-effective raw material for for scaffold production. But they did not mention the use of collagen from marine animals and fish, which became a substitute for animal collagen due to the possibility of encephalitis infection.
Conclusions should be shortened and concretized. In fact, the Conclusions look like a continuation of the review. Usually the conclusions do not contain references to the work of other authors.
Author Response
The aim of the work should be clearly stated. A review is usually part of a larger task. It is possible to create some new technology or improve the known ones.
Answer : We appreciate the valuable comments provided by the reviewer regarding the clear aim of this review. In accordance with the reviewer's suggestion, we have emphasized the aim of the review in the abstract of the revised manuscript.
(The aim of this review is to discuss the latest research trends in fibrous scaffold manufacturing methods, materials, drug loading methods with parameters information and therapeutic applications with the goal of contributing to the development of new technologies or improvements to existing ones.)
------------------------------
".....Furthermore, collagen can be easily extracted from natural sources such as bovine hide, pig skin, and human placenta, making it a readily available and cost-effective material for scaffold production". row 170-172
The authors, referring to a single source, identified the human placenta as a cost-effective raw material for for scaffold production. But they did not mention the use of collagen from marine animals and fish, which became a substitute for animal collagen due to the possibility of encephalitis infection.
Answer : We appreciate the reviewer's valuable comment regarding the additional source of collagen. Following the reviewer's suggestion, we have added sentences discussing the marine source of collagen on revised manuscript.
(In addition, marine organisms are being increasingly investigated as a potential source of collagen due to lower risk of transmissible diseases. Unfortunately, the fishing industry discards a significant amount of marine biomass, up to 85%, leading to environmental concerns. However, extracting collagen from these wastes can offer both environmental and economic benefits. Marine collagens have unique properties, such as biocompatibility, reduced zoonotic and immunological risks, and fewer religious restrictions.)
------------------------------
Conclusions should be shortened and concretized. In fact, the Conclusions look like a continuation of the review. Usually the conclusions do not contain references to the work of other authors.
Answer : We appreciate the reviewer's valuable comment regarding the conclusion part. In the revised manuscript the conclusion part is separated into “Future perspective and limitations”, and “conclusion”. In the revised “Future perspective and Limitation” section, we mentioned the limitations of therapeutic agent-loaded fibrous scaffolds, including the need to consider regulatory aspects, the system differences between humans and animals, and the necessity of considering mass production and reproducibility. We believe that the revised “Conclusion” part is properly shortened and concretized.
Reviewer 2 Report
Regarding the manuscript (pharmaceutics- 2357350) entitled:
“Therapeutic Agent-Loaded Fibrous Scaffolds for Biomedical Applications”
Comments to the Author
General comment
The manuscript describes the various methods for fabricating bioactive molecule-loaded fibrous scaffolds. I have some few comments to be considered before publication:
1. Previously published reviews like
https://www.ncbi.nlm.nih.gov/pmc/articles/PMC4549138/
https://www.mdpi.com/2079-6439/11/2/21
2. Abstract: It should show different sections covered in the review article.
3. Natural Materials for Fibrous Scaffolds: Chemical structure of each material should be added
4. Fabrication Methods: Diagram for each technique and parameters affecting the process should be added
5. Limitations of marketed products and future aspects should be presented as the final section.
Author Response
The manuscript describes the various methods for fabricating bioactive molecule-loaded fibrous scaffolds. I have some few comments to be considered before publication:
- Previously published reviews like
https://www.ncbi.nlm.nih.gov/pmc/articles/PMC4549138/
https://www.mdpi.com/2079-6439/11/2/21
Answer : We appreciate reviewer’s introduction of two excellent review papers. Each paper provides a comprehensive review of either electrospinning or other fabrication methods in details as follows:
Regarding the first review article titled “Nanofibrous scaffolds in biomedical applications”, this review article focuses on the applications and significance of electrospun nanofibrous scaffolds in various biomedical fields such as drug delivery and wound healing. The review also discusses the advantages and disadvantages of using electrospinning to fabricate nanofibrous scaffolds for biomedical applications. The article further examines the factors that influence drug distribution in electrospun nanofibrous scaffolds, with the goal of improving the therapeutic efficacy of these scaffolds in drug delivery and wound healing applications.
In contrast, our review covers the various materials, fabrication methods including electrospinning, and loading methods with parameters and considerations. In addition, biomedical application part is categorized on the purpose of the fibrous scaffolds.
Regarding the second review article titled “Nanofibres in Drug Delivery Applications”, this review article focuses on the potential of drug-loaded nanofibers for drug delivery and provides a detailed overview of various production methods. The review analyzes the relevant literature to explore the potential of nanofibrous-based pharmaceutical formulations. The article emphasizes the importance, versatility, and flexibility of nanofibers in creating medicines with diverse drug release kinetics.
Unlike the aforementioned, our review discusses materials, principles and detailed explanations of microfluidic fabrication, descriptions and differences in 3D printing techniques, and drug loading methods, aiming to decide the optimal approach.
---------------------------------------------------------------------------------------
- Abstract: It should show different sections covered in the review article.
Answer : We appreciate reviewer’s comment on the abstract. Our review aims to contribute to the development and improvement of regenerative medicine by providing a comprehensive discussion of materials, principles and differences of various methods for fabrication of fibrous scaffolds, drug loading methods, and applications categorized by their purposes. By adding the following sentences to the abstract, we believe that the aim and difference of this review is clarified further.
(The aim of this review is to discuss the latest research trends in fibrous scaffold manufacturing methods, materials, drug loading methods with parameters information and therapeutic applications with the goal of contributing to the development of new technologies or improvements to existing ones.)
---------------------------------------------------------------------------------------
- Natural Materials for Fibrous Scaffolds: Chemical structure of each material should be added
Answer : We appreciate the reviewer's valuable comment regarding the chemical structure of each material. In the revised manuscript, we added figure 1-4 showing chemical structure or illustrations of each materials.
----------------------------------------------------------------------------------
- Fabrication Methods: Diagram for each technique and parameters affecting the process should be added
Answer : We appreciate the reviewer's valuable comment regarding the parameters affecting the process. In the vised manuscript, we added diagram with figure 5 and 6 showing scheme of each method. In addition to that we added information of parameters and consideration not only for fabrication methods but drug loading methods with table 2 and table 3.
---------------------------------------------------------------------------------------
- Limitations of marketed products and future aspects should be presented as the final section.
Answer : We appreciate the reviewer's valuable comment on marketed product and future aspect. According to the reviewer’s comment, Limitations of marketed products and future aspect is added on the revised “6. Future perspective and limitations” parts .
Reviewer 3 Report
In this review article, the authors discussed about the therapeutic agent loaded fibrous scaffolds and their biomedical applications. The article is scholarly written, and the content is comprehensive and useful for the researchers working on fibrous biomaterials. So, I recommend for acceptance of the paper.
Author Response
In this review article, the authors discussed about the therapeutic agent loaded fibrous scaffolds and their biomedical applications. The article is scholarly written, and the content is comprehensive and useful for the researchers working on fibrous biomaterials. So, I recommend for acceptance of the paper.
Answer : We appreciate so much for your positive comments.
Round 2
Reviewer 2 Report
no comments